# Additional Global Climate Cooling by Clouds due to Ice Crystal Complexity

Emma Järvinen[1], Olivier Jourdan[2], David Neubauer[3], Bin Yao[4], Chao Liu[4], Meinrat O. Andreae[5,6], Ulrike Lohmann[3], Manfred Wendisch[7], Greg M. McFarquhar[8,9], Thomas Leisner[1], and Martin Schnaiter[1]

[1]Karlsruhe Institute of Technology, Institute of Meteorology and Climate Research, Karlsruhe, Germany
[2]Laboratoire de Météorologie Physique, Université Clermont Auvergne, OPGC, UMR/CNRS 6016, Clermont-Ferrand, France
[3]Institute of Atmospheric and Climate Science, ETH Zürich, Zürich, Switzerland
[4]Collaborative Innovation Center on Forecast and Evaluation of Meteorological Disasters, Nanjing University of Information Science and Technology, Nanjing 210044, China
[5]Biogeochemistry Department, Max Planck Institute for Chemistry, Mainz, Germany
[6]Scripps Institution of Oceanography, University of California San Diego, La Jolla, California, USA
[7]Leipzig Institute for Meteorology, University of Leipzig, Leipzig, Germany
[8]Cooperative Institute for Mesoscale Meteorological Studies, University of Oklahoma, Norman, OK
[9]School of Meteorology, University of Oklahoma, Norman, OK

**Correspondence:** Emma Järvinen (jarvinen@ucar.edu)

**Abstract.** Ice crystal submicron structures have a large impact on the optical properties of cirrus clouds and consequently on their radiative effect. Although there is growing evidence that atmospheric ice crystals are rarely pristine, direct in situ observations of the degree of ice crystal complexity are largely missing. Here we show a comprehensive in situ dataset of ice crystal complexity coupled with measurements of the cloud angular scattering functions collected during a number of observational, airborne campaigns at diverse geographical locations. Our results demonstrate that an overwhelming fraction (between 61 and 81%) of atmospheric ice crystals sampled in the different regions contain mesoscopic deformations and, as a consequence, a similar flat and featureless angular scattering function is observed. A comparison between the measurements and a database of optical particle properties showed that severely roughened hexagonal aggregates optimally represents the measurements in the observed angular range. Based on this optical model, a new parameterization of the cloud bulk asymmetry factor was introduced and its effects were tested in a global climate model. The modelling results suggest that due to ice crystal complexity, ice-containing clouds can induce an additional short wave cooling effect of $-1.12\,\mathrm{W\,m^{-2}}$ on the top-of-the atmosphere radiative budget that has not yet been considered.

## 1 Introduction

Atmospheric ice crystals exhibit considerable variability in growth habits (Heymsfield and Platt, 1984; Korolev et al., 1999; Lawson et al., 2006), which makes their representation in global and regional climate and numerical weather prediction models challenging. Moreover, laboratory observations and satellite retrievals have shown that ice crystal mesoscopic structures, such as surface roughness or other crystal deformations, which have been observed in various environmental conditions (Ulanowski

et al., 2006; Diedenhoven et al., 2012; Neshyba et al., 2013; Cole et al., 2014; Magee et al., 2014; Ulanowski et al., 2014), can further complicate their realistic representation. Ice crystal surface roughness was added as a new variable to models of ice particle optical properties (e.g. Macke et al., 1996; Yang and Liou, 1998; Baran et al., 2001; Baran and Francis, 2004; Sun et al., 2004; Yang et al., 2008; Baum et al., 2010; Platnick et al., 2017). Later, it was found that ice crystal parameterizations implementing roughened surfaces represent the measured optical properties, and especially the polarization effects of atmospheric ice clouds, more accurately than parameterizations based on a mixture of pristine ice crystals (Baran and Labonnote, 2006; Um and McFarquhar, 2007; Jourdan et al., 2010; Liu et al., 2014b; Yi et al., 2016; Tang et al., 2017). Currently, severely roughened aggregated ice crystals are assumed in remote sensing retrieval algorithms (Platnick et al., 2017), and it has been suggested to include this type of ice crystals in the radiative transfer algorithms of general circulation models (Yi et al., 2016). However, the observational justification of this approach is still lacking, mainly because sufficient evidence of frequent occurrence of roughened ice crystals in different types of ice-containing clouds has not been obtained yet.

Surface roughness changes the ice crystal single scattering properties significantly. Light scattering calculations have shown that, compared to pristine ice crystals, ice particles with roughened surfaces produce flat and featureless angular scattering functions that reveal a significantly elevated backward scattering and, therefore, a lower asymmetry factor compared to their smooth counterparts (Yang and Liou, 1998; Ulanowski et al., 2006; Baum et al., 2011; Yi et al., 2016). In situ observations at several geographical locations have given indications of low asymmetry factors in ice clouds in the range of 0.74 to 0.79 (Gerber et al., 2000; Gayet et al., 2006; Febvre et al., 2009; Jourdan et al., 2010). However, without simultaneous measurements of the ice particle surface roughness, it remains unclear if the measured low asymmetry factors of natural ice clouds are induced by this feature. In general, more measurements of the cloud asymmetry factor are needed, since a small change in the asymmetry factor can have significant consequences for the short wave cloud radiative effect (SWCRE). Yi et al. (2013) showed that assuming severely roughened ice crystals and, thus, lowering the cloud short wave (SW) asymmetry factors by values between 0.01 and 0.035, can cause an additional SW cooling of $1\text{-}2\,\mathrm{W\,m^{-2}}$ at the top-of-the atmosphere.

Recent developments in airborne in situ measurement techniques have enabled to directly measure ice crystal complexity at mesoscopic scales (here defined as structures in a scale between $100\,\mathrm{nm}$ to $10\,\mathrm{\mu m}$), which had previously been too small to be resolved from cloud particle imager measurements. The Small Ice Detector Mark 3 (SID-3) (Kaye et al., 2008) records the spatial distribution of coherent laser light scattered by individual ice crystals (examples of scattering patterns are shown in Fig. 1). The image texture of the resulting single-particle scattering patterns can be analysed to retrieve the so-called complexity parameter, $k_e$, that has proven to be a suitable proxy for the actual ice crystal mesoscopic complexity (Schnaiter et al., 2016). In this context, mesoscopic complexity comprises all crystal deformations (e.g., surface roughness, hollowness, and air inclusions), which result in the formation of speckles in the coherent light scattering by these particles. Since the SID-3 instrument does not discriminate between mesoscopic complexity and surface roughness, for the remainder of this paper the term *ice crystal mesoscopic complexity* is used instead of the more established term *ice crystal surface roughness*.

Here, the complexity analysis is applied to cloud chamber studies of simulated cirrus clouds and to globally distributed measurements from five airborne measurement campaigns conducted between 2011 and 2017 during spring and summer covering regions from the Tropics to the Arctic. The observations of the ice crystal mesoscopic complexity are linked to measurements of

the ice particle angular scattering function performed at various geographical locations in the southern and northern hemisphere with two polar nephelometers, the Particle Habit Imaging and Polar Scattering (PHIPS) probe, and the Polar Nephelometer (PN). In two cases the crystal complexity measurements and the angular scattering measurements were conducted simultaneously on the same ice particle populations. The measurement methods and locations are discussed in Sect. 2 and the results in Sect. 3. To assess the significance of the observations to the magnitude of the SWCRE, the measured cloud angular scattering function was parameterized and the new parameterization was tested in the ECHAM-HAM global climate model, and compared against results generated by the standard parameterization. The respective results of the simulations are discussed in Sect. 4.

## 2 Methods

### 2.1 Ice particle complexity analysis

The mesoscopic complexity of individual sub-$50\,\mu m$ ice particles was determined using the SID-3 instrument (Kaye et al., 2008), which records the spatial intensity distribution of coherent laser light scattered in the angular range of 7 to 23° as a two-dimensional (2-D) scattering pattern. Representative examples of respective scattering patterns are shown in Fig. 1. The crystal complexity is quantified from the 2-D scattering patterns using a grey-level co-occurrence matrix (GLCM) method (Lu et al., 2006). This approach was developed for industrial quality control of surface treatment processes, and was later adapted for analysis of complexity features of three-dimensional ice particles (Ulanowski et al., 2010, 2014; Schnaiter et al., 2016). A more detailed description of the analysis of ice crystal scattering patterns used in this study can be found in Schnaiter et al. (2016).

The GLCM analysis was performed only for scattering patterns that were well-illuminated and contained less than 15 % saturated pixels. To be consistent with laboratory studies by Schnaiter et al. (2016), the SID-3 camera gain settings were chosen between 175 and 195, and only images within a narrow mean brightness range between 10 and 50 were selected. These steps were taken to minimize image brightness biases on the GLCM analysis.

Although, the SID-3 has an open geometry to minimize artefacts due to ice particle shattering on the probe housing (McFarquhar et al., 2007; Cotton et al., 2010; Korolev et al., 2011), on some occasions shattering events are observed. 2-D scattering patterns from shattered particles can be distinguished from "real" ice particles by analysing the particle time-of-flight (TOF). A typical residence time in the $160\,\mu m$-diameter laser beam at an airspeed of $200\,m\,s^{-1}$ is $0.8\,\mu s$ that, divided by the $21\,ns$ clock cycle, corresponds a TOF of 38. In a shattering event, shattered ice crystal fractions pass the sensitive area of the laser beam with short-enough inter-arrival times, such that the electronics cannot resolve the individual pulses but instead a long TOF value is recorded. To exclude analysing 2-D scattering patterns belonging to shattered particles, the TOF was empirically limited to values below 350. This led to a removal of around 1 % of the 2-D scattering patterns with mean brightness ranges between 10 and 50 measured in high altitude clouds. In mixed-phase clouds a higher fraction of measured 2-D scattering patterns, between 7.5 % and 19 %, were excluded from analysis. The higher fraction of shattering in mixed-phase clouds can be explained by the presence of rimed particles (Jackson et al., 2014).

The result of the GLCM analysis is an optical complexity parameter, $k_e$, that covers values approximately between 4 and 6 depending on the degree of the actual ice crystals mesoscopic complexity. It was shown using both discrete dipole approximation light scattering calculations and cloud chamber simulations, that there is a correlation between the optical complexity parameter $k_e$ and the physical surface roughness in the range from 0.1 to about 1 µm (Schnaiter et al., 2016). Therefore, it is justified to use $k_e$ as a measure of ice crystal mesoscopic complexity. However, it should be noted that $k_e$ is an optical parameter and cannot be directly translated into a physical complexity measure or to a distortion parameter used in optical particle models.

## 2.2 In situ measurements of the angular scattering function

The angular scattering functions of individual ice particles at 532 nm wavelength were measured with the PHIPS aircraft probe (Abdelmonem et al., 2016; Schnaiter et al., 2018). PHIPS is capable of measuring the angular scattering function of individual particles from 18 to 170° with a repetition rate as high as 13 kHz. The particle size range covered is from 10 µm to approximately 1 mm in diameter. Simultaneously, a stereoscopic image is taken for a sub-sample of particles. Examples of PHIPS images of tropical ice particles are presented in Fig. 1. Before analysis, particles corresponding to shattering events were removed by calculating particle inter-arrival times and removing particle pairs with inter-arrival times less than 1 ms.

The angular scattering measurements at 804 nm wavelength were performed with the PN (Gayet et al., 1997; Crépel et al., 1997). The PN measures the angular scattering coefficients of particle populations by integrating the measured signals of each detector over a period selected by the operator (typically 100 ms). The particle size range is from few micrometres to 1 mm. The scattering angles of PN cover values from 15° to 162° with a resolution of 3.5°. It is not possible to correct the PN data for shattering artefacts, but it has been estimated that shattering artefacts contribute less than 25% to the total extinction signal (Mioche et al., 2017).

## 2.3 Cloud chamber experiments

Cloud chamber experiments were performed to study the effect of growth conditions to the ice crystal mesoscopic complexity. These experiments were performed at the AIDA cloud simulation chamber of Karlsruhe Institute of Technology during a series of **R**ough **ICE** (RICE) experiments. A general description of the AIDA facility and instrumentation can be found in several publications (e.g. Möhler et al., 2005; Wagner et al., 2011; Schnaiter et al., 2012) and the detailed description of the RICE experiments can be found in Schnaiter et al. (2016). Here, we compare field results of ice crystal complexity measurements to four laboratory experiments from Schnaiter et al. (2016) that represent simulations of pristine, pristine to medium complex, medium complex to severe complex, and severe complex ice crystals (Table 1).

Each simulation experiment started with a pre-cooled and pre-humidified chamber (temperature of -50°C and relative humidity with respect to ice, $RH_{ice}$, of 100%). Before the experiment, the chamber was filled with either sulphuric acid solution droplets for simulations of homogeneous freezing, or with soot aerosol particles for simulations of heterogeneous deposition mode freezing. In the first experiment phase the aerosol was activated by expanding the chamber volume through evacuation. During the initial activation, the ice particle growth conditions cannot be controlled and, therefore, a subsequent sublimation is

needed to remove any morphological features related to the initial growth. In the second phase, the ice particles were reduced in size before they were, in the third experiment phase, re-grown at a stable ice supersaturation. During the re-growth period the ice particles were analyzed in terms of their mesoscopic complexity.

## 2.4 Sampled clouds and definitions

Field measurements of ice crystal mesoscopic scale complexity were performed between 2011 and 2017 in the Mid-latitude Airborne Cirrus Properties Experiment (MACPEX) (Jensen et al., 2013), the Mid-Latitude Cirrus (ML-CIRRUS) campaign (Voigt et al., 2017), the Aerosol, Cloud, Precipitation, and Radiation Interactions and Dynamics of Convective Cloud Systems Cloud processes of the main precipitation systems in Brazil: A contribution to cloud resolving modeling and to the GPM (GlobAl Precipitation Measurement) (ACRIDICON-CHUVA) campaign (Wendisch et al., 2016), the Radiation-Aerosol-
Cloud Experiment in the Arctic Circle (RACEPAC) campaign (Costa et al., 2017), and the Arctic CLoud Observations Using airborne measurements during polar Day (ACLOUD) campaign (Wendisch and et al., 2018). The field measurements of angular scattering functions were performed between 1998 and 2017 in the Interhemispheric differences in Cirrus properties from Anthropogenic emissions (INCA) project (Shcherbakov et al., 2005), the mid-latitude campaign CIRRUS'98 (Jourdan et al., 2003), the Arctic Study on Tropospheric Aerosol and Radiation (ASTAR) campaign (Jourdan et al., 2010), the Contrail and
Cirrus Experiments (CONCERT) 1 and 2 (Chauvigné et al., 2018), the tropical campaign ACRIDICON-CHUVA, the Airborne Research Instrumentation Testing Opportunity (ARISTO2017), the Arctic campaign ACLOUD and, the Southern Ocean Clouds, Radiation, Aerosol Transport Experimental Study (SOCRATES).

In this paper the microphysical and optical properties of ice particles in high altitude clouds, in boundary layer stratocumulus clouds, and in one nimbostratus cloud are reported. The temperature ranges covered in each campaign are shown in Table
2. High altitude clouds were sampled in the tropical campaign ACRIDICON-CHUVA, in the mid-latitude campaigns ML-CIRRUS, MACPEX, ARISTO 2017, CIRRUS'98 and CONCERT, as well as in the Southern Ocean campaign SOCRATES, and in the northern and southern hemispheric campaign INCA. From these observations, only segments in fully glaciated parts were selected for the analysis. This included measurements above -40°C and, therefore, in this study a more general term ice clouds instead of cirrus clouds is used.

Boundary layer stratocumulus clouds were sampled in the Arctic campaigns RACEPAC and ACLOUD, and in the Southern Ocean campaign SOCRATES. Different approaches were applied to select ice particles for the analysis. For the SID-3, a manual inspection of the single particle 2-D scattering patterns was performed for the RACEPAC and ACLOUD flights, where ice was observed. The complexity analysis was performed only for scattering patterns classified manually as ice. To calculate a representative angular scattering function for boundary layer stratocumulus ice particles, the PHIPS single particle
angular scattering functions from the ACLOUD and SOCRATES campaigns were first analyzed for their shape. The shape of the rainbow feature, that is the slope between 106° and 138°, was used to discriminate between liquid droplets and ice particles. Only particles that were classified as ice using this algorithm were included in the analysis. In the Arctic campaigns, all the analyzed ice particles were measured in a mixed-phase environment. Therefore, the term Arctic mixed-phase ice is used to label the PHIPS measurements. The PHIPS measurements in the Southern Ocean campaign SOCRATES includes ice

particles sampled both in high altitude clouds and in boundary layer stratocumulus (mixed-phase) clouds. In this paper, one representative angular scattering function for Southern Ocean SOCRATES campaign is shown.

An Arctic mixed-phase nimbostratus cloud was sampled during the ASTAR campaign. To retrieve a representative ice particle angular scattering function for Arctic ice particles, principal component analysis (Jourdan et al., 2003) was performed on the PN data measured at the glaciated top of this system. Since the cloud top was almost fully glaciated (Jourdan et al., 2010), the measurements are labeled in this paper as Arctic ice cloud.

## 2.5 Description of the ECHAM-HAM model.

In our study, we used the ECHAM6.3-HAM2.3 global aerosol-climate model (based on Neubauer et al. (2014) with modifications). A 10-year simulation with 1.9° x 1.9° horizontal resolution with 47 vertical levels was conducted from 2003 to 2012 after 3 months of spin-up time. The meteorology is nudged to ERA-Interim data (Dee et al., 2011) and sea surface temperature and sea ice cover were taken from observations. In the model the radiative transfer is computed for 14 short wave bands (and 16 longwave bands) (Pincus and Stevens, 2013). A competition between homogeneous and heterogeneous nucleation and pre-existing ice crystals (Kuebbeler et al., 2014; Gasparini and Lohmann, 2016) is considered. Enhancements in the vertical velocity over orography (Joos et al., 2008) are accounted for in the formation of cirrus clouds.

The ECHAM-HAM model is used to calculate the SWCRE of the ice clouds, which is computed online by calling the radiation subroutine twice. The first call is with clouds (all-sky) and the second call is without clouds (clear-sky) in the atmosphere. The first call uses the standard model parameterization for the short wave asymmetry factors of ice clouds. The radiative fluxes from this call to the radiation subroutine are used to advance the model simulations. The cloud radiative effects are computed as the difference between the all-sky minus the clear-sky fluxes. To estimate the change in SWCRE by changing the short wave asymmetry factors of ice clouds an additional (third) call to the radiation subroutine is conducted. The additional (diagnostic) call to the radiation subroutine is identical to the first call except for using the new parameterization for the short wave asymmetry factors of ice clouds. The radiative fluxes from this additional call are only diagnostic. The SWCRE using the new parameterization for the short wave asymmetry factors of ice clouds is computed from the difference in SW radiative flux at the top of the atmosphere from the additional call and the cloud-free SW radiative flux at the top of the atmosphere.

## 3 In situ measurements

### 3.1 Globally distributed in situ observations of ice crystal mesoscopic complexity

The tracks of the measurement flights, where ice crystal mesoscopic complexity was studies, are shown in Fig. 2. The southernmost dataset of tropical ice clouds, collected during ACRIDICON-CHUVA campaign, consists mainly of measurements in anvil cirrus, but includes also two cases of synoptic cirrus. The dominant ice crystal habits in the anvil cirrus were found to be plates and aggregates of plates, whereas synoptic cirrus was composed of bullet rosettes and columnar ice crystals (examples of ice particles in a tropical in situ cirrus can be found in Fig. 1). The observations of the crystal habits agree with previous

observations in convective and synoptic systems (McFarquhar and Heymsfield, 1996; Heymsfield et al., 2002; Connolly et al., 2005; Lawson et al., 2006). In contrast to the tropical cirrus, the ice crystals measured in the ML-CIRRUS campaign were formed in more moderate updrafts in synoptic systems, such as warm conveyor belts or in the jet stream. During MACPEX, the second campaign in mid-latitudes analyzed here, dominant cirrus types were either anvil or jet stream cirrus associated with spring storm systems (Schmitt et al., 2016b). The northernmost campaigns targeted springtime Arctic boundary layer stratocumulus clouds from northern Canada (RACEPAC) and from Svalbard, Norway (ACLOUD). In ACLOUD, the ice crystals were found at temperatures between -3°C and -10°C, where the most common ice crystal shapes were (hollow) rimed needles or plates (Schnaiter et al., 2018).

Only ice crystals in the sub-50 μm size range were selected from the data obtained during these campaigns. In this size range, ice particles are single crystals (Schmitt et al., 2016a) and, therefore, complexity is observed in the mesoscopic-scale and is not caused by aggregate structures. Based on laboratory calibrations (Schnaiter et al., 2016) the measured ice crystals can be divided into pristine ($k_e < 4.6$) and complex ($k_e \geq 4.6$). Statistical analysis of the single particle complexity parameters measured in the different campaigns are shown in Fig. 3. This analysis reveals that a majority, between 61 and 81%, of the ice crystals with sizes below 50 μm, can be classified as complex with median complexity parameters above the defined threshold of $k_e \geq 4.6$. In spite of the obvious differences in the ice crystal habits due to the different formation pathways, the median complexity parameters have similar values in all campaigns. The maximum difference in the median complexity parameter was found 0.23 that roughly corresponds to a change of 0.05 in distortion parameter ($\sigma$) (Schnaiter et al., 2016) or 0.04 μm in physical surface roughness (Lu et al., 2006).

Even though the method is limited to the study of mesoscopic scale complexity of small ($<50$ μm) ice particles, it can be postulated that the results give indications also for the structural complexity of ice particles larger than 50 μm. Larger ice particles are frequently aggregates, composed of small single habits whose mesoscopic scale complexity can be measured (Schmitt et al., 2016b). It can be assumed that an aggregated ice crystal has the same or even higher degree of mesoscopic scale complexity as the single habits composing it and, therefore, the asymmetry factor of aggregated crystals is similar or lower than that of the component particles (Yang et al., 2013; Um and McFarquhar, 2009). For example, light scattering calculations have shown that the scattering properties of aggregated hexagonal ice crystals differ only little (around 0.3% at 550 nm) from those of their component particles (Um and McFarquhar, 2009).

### 3.2 Comparison of the field observations to laboratory simulation experiments

In Fig. 3, the atmospheric measurements are compared to four laboratory cirrus cloud experiments performed in the AIDA cloud chamber. In the cloud chamber experiments ice crystals were nucleated either homogeneously or heterogeneously at temperatures around -50°C, with a subsequent growth at a defined ice saturation ratio ($S_{ice}$) ranging from near ice saturation to 30% ice supersaturation (Schnaiter et al., 2016). The homogeneous freezing case (AIDA hom.) resulted in the highest degree of mesoscopic scale complexity (median $k_e$ of 5.33) even at moderate growth conditions ($S_{ice}$ of 1.1) whereas the degree of mesoscopic scale complexity in the case of heterogeneous freezing (AIDA het.) was dependent on the ice supersaturation ratio during crystal growth. The measured median $k_e$ values were 4.91, 4.68 and 4.22 for experiments where the crystal growth

took place at 30%, 20% and 1% supersaturated conditions, respectively. The difference in the physical surface roughness, as defined by Lu et al. (2006), between the AIDA het. 30% and AIDA het. 1% experiments would roughly be $0.12\,\mu m$. A similar enhancement in mesoscopic scale complexity in homogeneously formed ice crystals has previously been found in mid-latitude cirrus (Ulanowski et al., 2014), and can be partly explained by an increased stacking disorder of homogeneously nucleated
ice crystals (Malkin et al., 2012). The median and the interquartile range of the $k_e$ from the field observations agree best with the laboratory simulations of heterogeneous freezing where ice crystals were grown at relatively high $S_{ice}$ of 1.3 (Fig. 3). However, it has to be taken into account that in the atmosphere ice crystals can undergo several growth and sublimation cycles that contribute to the formation of additional crystal complexity after the initial growth (Magee et al., 2014; Chou et al., 2018).

### 3.3    Measurements of the angular scattering function

Our field results show that the degree of ice crystal mesoscopic complexity is always above the threshold value of 4.6, and shows less variation with geographical locations than the variations observed in the laboratory simulations. However, for estimating the ice cloud radiative effect it is crucial to understand how this microphysical observation affects the radiative properties of cirrus and mixed-phase clouds. Fig. 4 shows field and laboratory measurements of volumetric angular scattering functions at two solar wavelengths for a particle size range from $10\,\mu m$ to $1\,mm$ in diameter. Each function represents the median over a
whole campaign or over one geographical location. The measured angular scattering functions are flat and featureless. Studies with optical particle models (Doutriaux-Boucher et al., 2000; C-Labonnote et al., 2001; Baum et al., 2010; Jourdan et al., 2010; Yang et al., 2013; Liu et al., 2014a, b; Letu et al., 2016; Tang et al., 2017) show that the flattening of the angular scattering function at the sideward angles can be reproduced by ice particles with a high degree of crystal complexity, which is in accordance with our observations. More importantly, the ensemble angular scattering functions at both solar wavelengths are
almost identical irrespective of the geographical location. Although ice crystal habits differ significantly in convective outflows, in situ cirrus or in boundary layer stratocumulus clouds, this shows that their angular scattering behaviour is governed by the mesoscopic features of the crystals.

### 3.4    Comparison of the measured angular scattering functions to a light scattering database

The measured angular scattering functions at the two wavelengths were compared to theoretical phase functions for different
habits calculated using the database of Yang et al. (2013). In accordance with our observations, only severely roughened habits were considered in the theoretical calculations. For generation of the theoretical phase functions a representative ice particle size distribution from the ACRIDICON-CHUVA campaign was used. The size distribution was determined from the PHIPS images by analyzing the maximum dimension of each imaged ice particle using an algorithm developed by Schön et al. (2011). Furthermore, the sensitivity of the theoretical phase function to the assumed size distribution was investigated and it was
found that the shape of the phase function was insensitive to small changes in the median diameter. Figs. 5 and 6 show the measured and normalized volumetric angular scattering functions for 532 and $804\,nm$ and the theoretical phase functions for nine different habits.

Based on the comparison, the severely roughened column aggregate model was found to best represent the measurement at both wavelengths. At $532\,\mathrm{nm}$ the theoretical calculations agree with the measurements over the whole measurement range, whereas at $804\,\mathrm{nm}$ the model predicts slightly higher intensity in the sideward angles between $57°$ and $126°$ but is within the measured interquartile range (Fig. 6). The calculated root mean square errors (RMSE) between the severely roughened column aggregate model and the mean of the measurements are the lowest (0.0017 and 0.0014 for 532 and 804 nm, respectively) compared to the other models (RMSEs between 0.0022 and 0.0111 for 532 nm, and 0.0037 and 0.0208 for 804 nm). At the angles around exact-backscattering the severely roughened column aggregate model predicts a relatively flat behaviour. However, recent modelling studies have indicated that the scattering intensities around exact backscattering angles should be enhanced due to coherent scattering (e.g. Zhou, 2018). Although this effect can be important for lidar applications, it does not significantly affect the redistribution of the energy in the scattering process and, thus, the magnitude of the asymmetry factor. Furthermore, comparisons of satellite retrievals of cloud polarization properties with light scattering simulations have shown that optical particle models using severely roughened crystals yield the best agreement (Baum et al., 2011; Yang et al., 2013; Tang et al., 2017) and the current MODIS retrievals are based on the same optical particle model of severely roughened hexagonal aggregates that is used here (Platnick et al., 2017).

## 4   Estimating the effect of the observed mesoscopic scale complexity to SWCRE

An important consequence of severely roughened and complex ice crystals is that the cloud asymmetry factor in the solar spectral range is lowered compared to pristine ice crystals (e.g. Macke et al., 1996; Yang and Liou, 1998; Liou et al., 2000; Baum et al., 2010, 2011; Baran, 2012; Diedenhoven et al., 2012; Yang et al., 2013). For example, the severely roughened hexagonal aggregate model has relatively low asymmetry factors of 0.750 and 0.754 for $532\,\mathrm{nm}$ and $804\,\mathrm{nm}$, respectively. To understand the relevance of our observations for climate projections, the effect of the observed decrease in the cloud asymmetry parameter on the SWCRE was estimated by newly parameterizing the SW asymmetry factors using the optical model with the best fit to our measurements in the ECHAM-HAM global climate model. The current optical parameterization in the ECHAM-HAM model is calculated based on spherical particles using Mie-theory with the exception that the asymmetry factors are scaled down to be more representative for aspherical ice particles. The steps to retrieve the new parametrization are discussed in Sect. 4.1. The sensitivity of a global climate model to the ice particle surface roughness has already been tested in the study of Yi et al. (2013), where the difference in the SWCRE was calculated for assuming first completely smooth and later severely roughened ice particles. Here, we compare the existing standard parameterization of SW asymmetry factors to our new parametrerization and, in this way, estimate the possible impact of of the observed ice crystal mesoscopic scale complexity to the SWCRE.

## 4.1 Derivation of the new parameterization of the short wave asymmetry factor for the ECHAM-HAM model and comparison with the standard parameterization.

Fig. 4 showed that the observed high degree of mesoscopic scale complexity dominates the angular scattering function over the ice crystal shape and a uniform angular scattering function is observed at two wavelengths (532 and 804 nm). Therefore, it is justified to use a single-habit optical ice particle model assuming severely roughened surfaces to compute the bulk optical properties of ice clouds. It was found that the severely roughened column aggregate model showed the best fit of the atmospheric measurements performed at both wavelengths. At 804 nm the model disagreed slightly with the measurements at the sideward angles (Fig. 4). This disagreement indicates that either the severely roughened column aggregate model does not accurately represent the spectral dependence of the asymmetry factors, or could also be related to systematic measurement uncertainties caused by using different measurement systems. However, since we only have information on the ice particle angular scattering properties at two wavelengths at the moment, only one optical particle model is used to parameterize the asymmetry factors.

Gamma particle size distributions with a variance of 0.1 were used to calculate the bulk asymmetry factors at each wavelength for different effective radii ranging from 4 to 124 μm. The comparison of the standard parameterization in ECHAM-HAM for the SW asymmetry factors and the new parameterization using the severely roughened column aggregate model is shown for selective wavelength bands in Fig. 7. As expected, the new asymmetry factors are lower than what is assumed in the standard parameterization, except for the 3.47 μm band. Another consequence of the particle roughening is that the size dependence of the asymmetry factor becomes weaker and for sub-micron wavelength bands (0.23, 0.4 and 0.7 μm) almost no size dependence is observed. It seems that due to mesoscopic complexity the ice cloud asymmetry factors are not so sensitive to habits or particle size, whereas previous studies on smooth ice crystal have shown that different parameterizations using different habits or habit mixtures can cause a significant variance in the asymmetry factor by 0.07 in the wavelength band of 0.25 to 0.69 μm (McFarquhar et al., 2002). This variance becomes especially significant for small ice particles, with effective radius below 20 μm, where also the largest uncertainty in the exact particle shape exists.

## 4.2 Influence of ice crystal mesoscopic complexity to the cloud shortwave radiative effect

The change in the global SWCRE after applying the new parameterization to all ice clouds (cirrus and mixed-phase) is shown in Fig. 8. The global mean change in the SWCRE is $-1.12 \, \mathrm{W\,m^{-2}}$, but regionally it can be as large as $-8 \, \mathrm{W\,m^{-2}}$. If the new parameterization is applied only for cirrus clouds, the mean change in the SWCRE is slightly lower, $-1.00 \, \mathrm{W\,m^{-2}}$. Therefore, the change in the asymmetry factor mostly affects the cirrus SWCRE and, also, the largest effect is found in the tropical regions where also the cirrus occurrence is the highest (e.g. Sassen et al., 2008). Even though the change in the global SWCRE is small compared to the global mean SWCRE of all clouds of about $-50 \, \mathrm{W\,m^{-2}}$ (Boucher et al., 2013) or to the global mean SWCRE of ice clouds of about $-16.7 \pm 1.7 \, \mathrm{W\,m^{-2}}$ (Hong et al., 2016) it is approximately one fourth of the global mean cirrus SWCRE of $-4 \, \mathrm{W\,m^{-2}}$ (Gasparini and Lohmann, 2016) and comparable to the total direct radiative effect of aerosols ($-2.1 \pm 0.7 \, \mathrm{W\,m^{-2}}$) (Lacagnina et al., 2017). The enhanced SW cooling might have important implications on understanding the cirrus CRE not only on global but also on regional scale. For example, the increased reflectivity might change our assessment of the

sign of the cloud radiative effect by thin cirrus. So far, thin cirrus has been considered to have a modest but positive cloud radiative effect (around $0.7\,\mathrm{W\,m^{-2}}$) (McFarquhar et al., 2000), but our results suggest that this needs to be reconsidered.

## 5   Conclusions

Although current satellite retrievals and a growing number of climate models have already started using optical parameteri-zations assuming severely roughened ice crystals to reproduce the observed flat angular scattering function of ice particles, this study gives the first direct observational evidence of ice crystal complexity and links it to an angular scattering function with low asymmetry factor. The results presented here show that optical models assuming severe roughness can represent the angular scattering function in many geographical locations with sufficient accuracy. Thus, based on observational evidence, the current uncertainty in the degree of surface roughness of natural ice particles (Cole et al., 2014) can significantly be reduced. Moreover, since the ice particle angular scattering functions did not vary significantly between different geographical locations, the modelling efforts of ice particle optical properties in future weather forecast and climate models will be simplified.

In situ measurements of the mesoscopic complexity using the SID-3 instrument showed that the majority of measured ice crystals can be classified as complex. The limitation of this method is that only small ($<50\,\mathrm{\mu m}$) ice crystals can be analyzed, and no direct evidence of the mesoscopic complexity of larger ($>50\,\mathrm{\mu m}$) ice crystals can be obtained. However, the angular scattering measurements show indirect evidence that larger ice particles are also likely complex. This can be seen from Fig. 4 by comparing the angular scattering functions of laboratory generated sub-$50\,\mathrm{\mu m}$ ice particles and that of natural ice particles. Although the field observations include a wider size range of ice crystals from few tens of microns up to a millimetre, no difference can be observed in the shape-sensitive sideward angular scattering behaviour between laboratory generated single habits and field observations.

Our modelling results showed that the observed ice particle mesoscopic scale complexity can significantly affect the SWCRE due to lowering of the cloud asymmetry factor. The magnitude of the change in the SWCRE of $-1.12\,\mathrm{W\,m^{-2}}$ is significant, but in order to estimate the role of ice crystal mesoscopic scale complexity for climate projections, future simulations with severely roughened ice crystals in a warmer climate are needed.

*Data availability.* The SID-3 complexity analysis results from ML-CIRRUS and ACRIDICON-CHUVA are available from the HALO database (https://halo-db.pa.op.dlr.de). The PHIPS data and SID-3 data from other campaigns are available upon request from Martin Schnaiter (martin.schnaiter@kit.edu).

*Author contributions.* E.J. and M.S. collected and analysed the SID-3 and PHIPS data from aircraft and AIDA campaigns. O.J. provided the PN data. B.Y. and C.L. performed the optical modelling for retrieval of the asymmetry factors and created the new parameterization of the asymmetry factors for the ECHAM-HAM model. D.N. and U.L. performed the ECHAM-HAM model runs. E.J., M.S., O.J., D.N., C.L.,

30  M.A., U.L., M.W., G.M. and T.L. were involved in the scientific interpretation and discussion. E.J. wrote the manuscript with help from O.J. and D.N. All commented on the paper.

*Competing interests.* The authors declare that they have no competing financial interests.

*Acknowledgements.* We gratefully acknowledge Steffen Münch for implementing the used cirrus scheme into ECHAM-HAM. We would also like to thank all participants of the field studies for their efforts, in particular the technical crews of the HALO, AWI Polar 6 and Polar
5  2, TBM700, DLR Falcon, NASA WB-57 and NSF G-V. This work has received funding from the Helmholtz Research Program Atmosphere and Climate, the German Research Foundation (DFG grants SCHN 1140/1-1, SCHN 1140/1-2, SCHN 1140/3-1) within the DFG priority program 1294 (HALO), the German Max Planck Society, the CNES (Centre National des Etudes Spatiales) and The Centre National de la Recherche Scientifique – Institut National des Sciences de l'Univers (CNRS-INSU) within the Expecting EarthCare Learning from A-Train (EECLAT) project (contract n°4500054452 BCT_69 2017), the Swiss National Supercomputing Centre (CSCS, project ID s652), the Swiss
10  National Science Foundation (project number 200021_160177) and the National Natural Science Foundation of China (grant no. 41571348) and by the United States National Science Foundation grands 1660544, 1628674 and 1762096. We gratefully acknowledge the National Science Foundation (NSF) for providing access to the HIAPER aircraft during the ARISTO 2017 project and the support from DFG within the Transregional Collaborative Research Center (TR 172) "Arctic Amplification: Climate Relevant Atmospheric and Surface Processes, and Feedback Mechanisms (AC)[3]" for providing access to the AWI Polar-6 aircraft during the ACLOUD project. The ECHAM-HAMMOZ
15  model is developed by a consortium composed of ETH Zürich, Max Planck Institut für Meteorologie, Forschungszentrum Jülich, University of Oxford, the Finnish Meteorological Institute, and the Leibniz Institute for Tropospheric Research, and managed by the Center for Climate Systems Modeling (C2SM) at ETH Zürich.

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

## Laboratory Produced Ice Particles

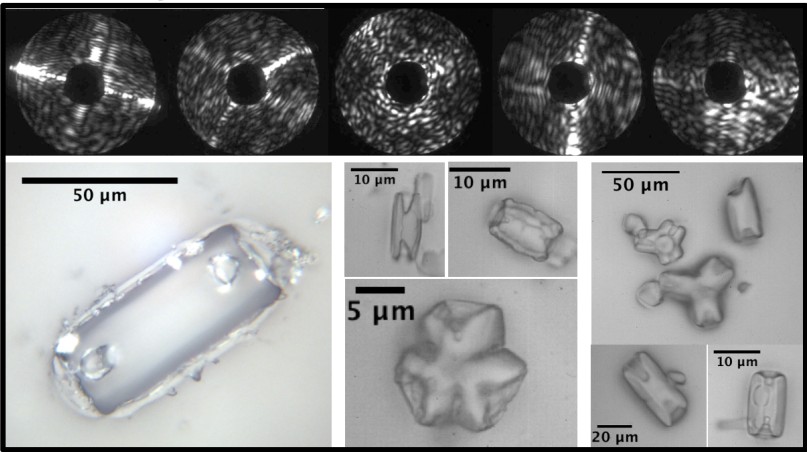

## Ice Particles in a Tropical Cirrus

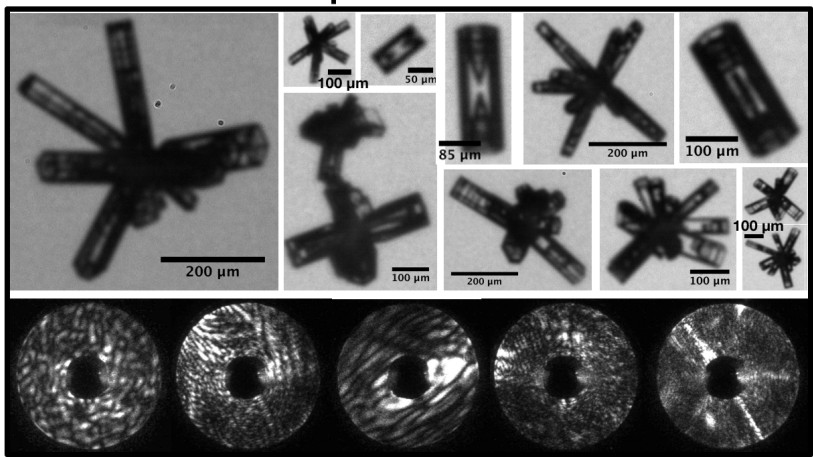

**Figure 1.** Hexagonal ice crystals and 2D diffraction patterns measured in laboratory simulations at -50°C (upper panel) and in tropical cirrus at -60°C (lower panel). The microscopic images of the laboratory produced ice crystals are from ice crystal replicas and the tropical cirrus ice particles were imaged in flight using bright field microscopy. The 2D diffraction patterns were measured simultaneously from the same particle population.

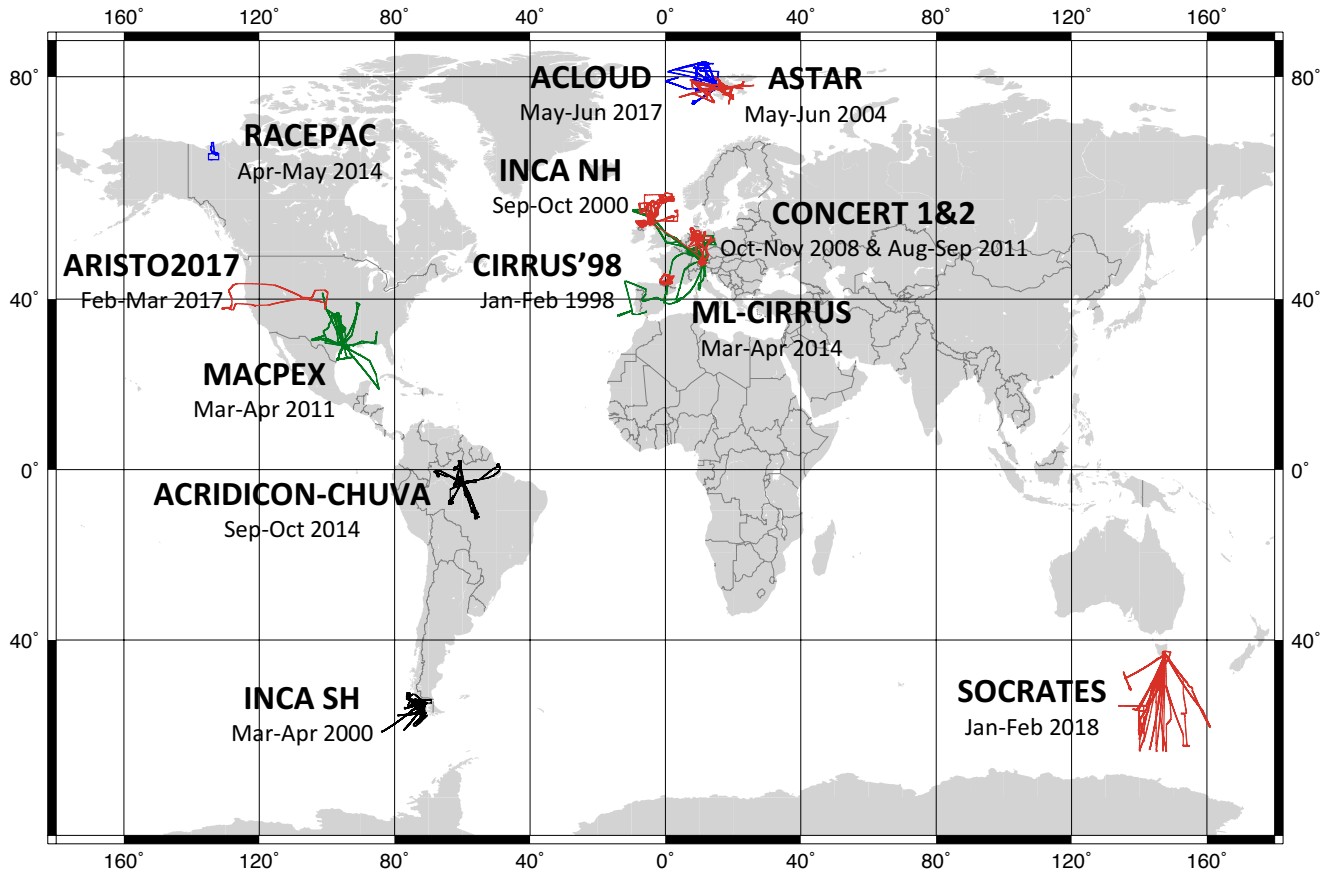

**Figure 2.** Flight trajectories of all campaigns included in this study. Trajectories of the campaigns where ice crystal mesoscopic scale complexity was investigated are marked with black, purple and blue matching the colours used in Fig. 3. Trajectories of the campaigns where only angular scattering function was measured are marked with red. Simultaneous mesoscopic scale complexity measurements and angular scattering measurements were performed in ACRIDICON-CHUVA and ACLOUD campaigns.

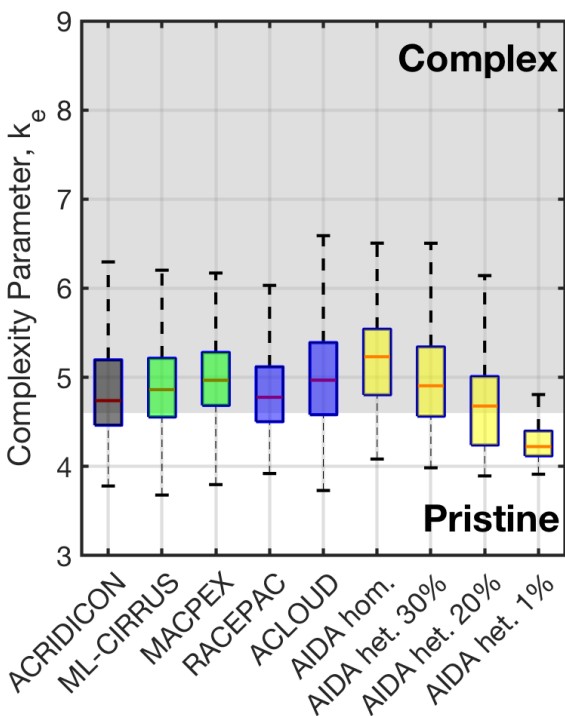

**Figure 3.** Statistical analysis of ice crystal complexity from all measured ice particles in the aircraft campaigns and from four AIDA cloud chamber simulation experiments. The box edges represent the 25 and 75% quartiles and the dashed lines the 5 and 95% quartiles. The red lines represent the median values. The grey area indicates the range of the complexity parameter in which the ice crystals are characterized to be complex. The median complexity parameters were found to be 4.74 in ACRIDICON-CHUVA, 4.86 in ML-CIRRUS, 4.97 in MACPEX, 4.78 in RACEPAC and 4.97 in ACLOUD.

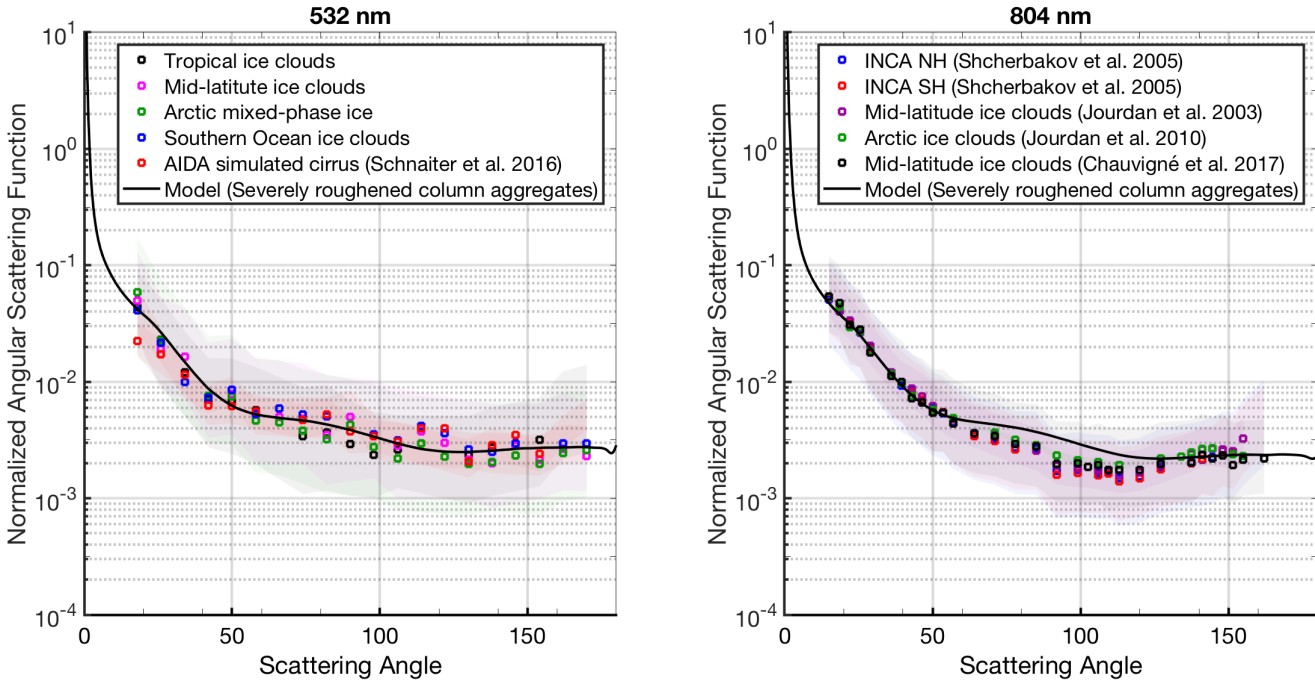

**Figure 4.** Ensemble angular scattering functions of natural and laboratory generated ice particles measured at two wavelengths. Each function represents the median angular scattering function over a single campaign and is normalized to the total intensity between 18° and 170°. The shaded area represents the interquartile range. The measurements at 532 nm in tropical cirrus, Arctic boundary layer stratocumulus clouds as well as the measurements in laboratory-simulated cirrus were gathered together with the complexity measurements in the ACRIDICON-CHUVA, ACLOUD and AIDA campaigns. The mid-latitude and Southern Ocean measurements at 532 nm were measured during the ARISTO campaign in 2017 and during the SOCRATES campaign, respectively. The measurements at 804 nm were measured between 1997–2011 in the CIRRUS'98, INCA, ASTAR and CONCERT aircraft campaigns.

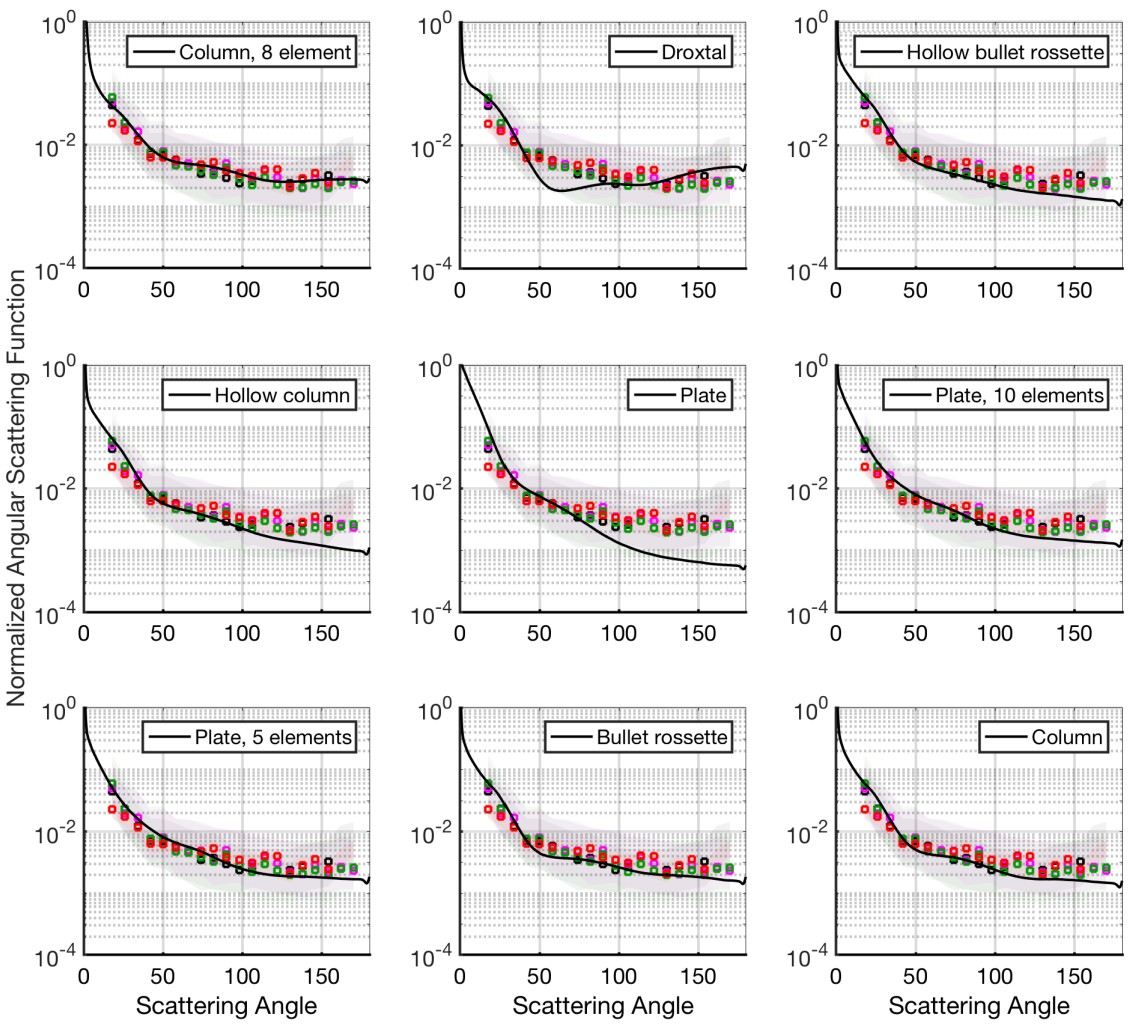

**Figure 5.** A comparison of the measured angular scattering functions at 532 nm (data from first panel of Fig. 4) and theoretical phase functions for different habits calculated using the database of Yang et al. (2013) and assuming a size distribution as measured during the ACRIDICON-CHUVA campaign. All calculations were performed assuming severely roughened surfaces. Both the measurements and the model results are normalized to the total intensity between 18° and 170°.

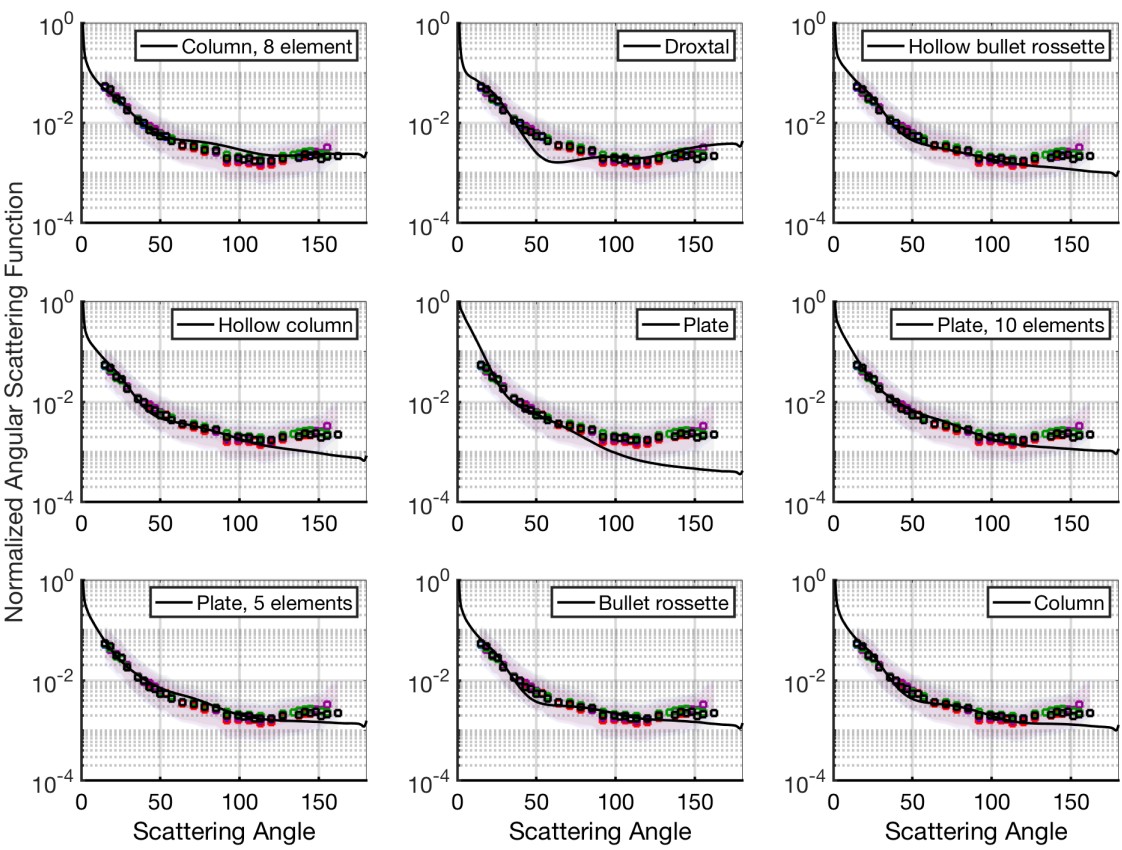

**Figure 6.** Same as Fig. 5 but now showing the comparison for PN measurements and for different optical models calculated for 804 nm. All calculations were performed assuming severely roughened surfaces. Both the measurements and the model results are normalized to the total intensity between 15° and 155°.

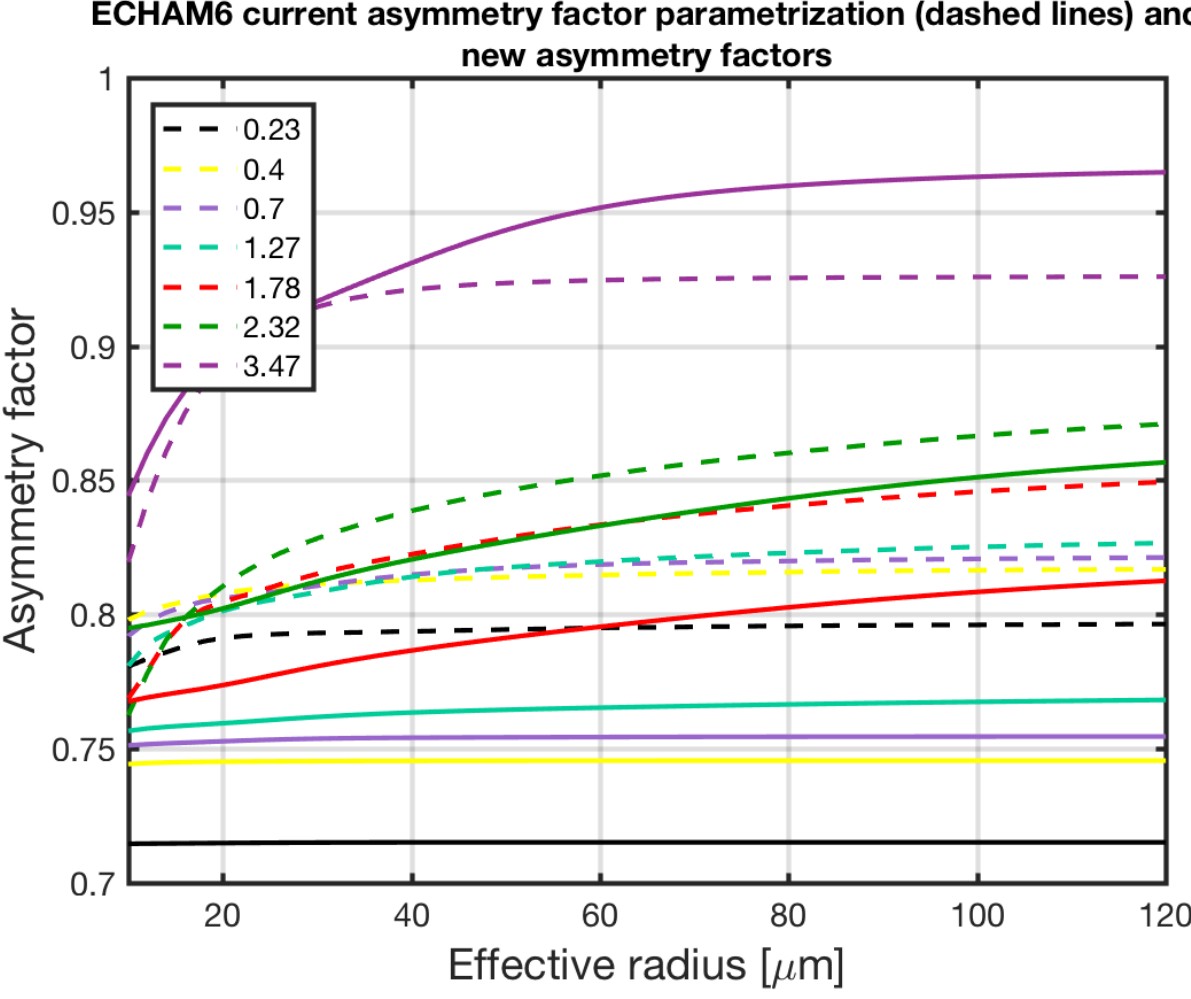

**Figure 7.** Comparison of the standard parameterization in ECHAM-HAM of the asymmetry factor of ice particle with different effective radius (dashed lines) and the new parameterization using severely roughened column aggregates (solid lines) for different wavelength bands. The wavelength bands are named with the band effective wavelength.

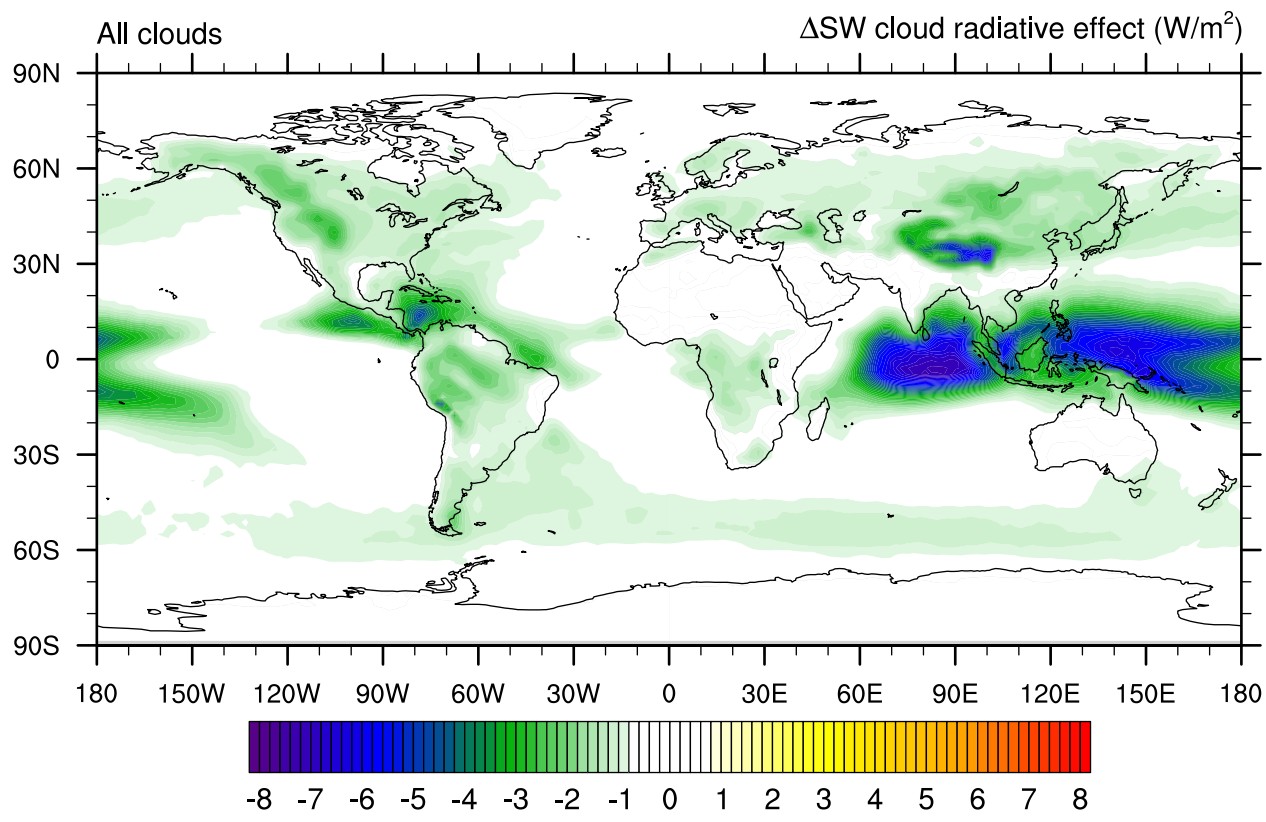

**Figure 8.** The global change in the shortwave cloud radiative effect predicted by the ECHAM-HAM model when the standard parameterization of the short wave asymmetry factor is substituted by the parameterization using severely roughened ice particles. In this simulation the new short wave asymmetry factors were applied to all ice particles both in cirrus and in mixed-phase clouds.

**Table 1.** Summary of the AIDA experiments shown in Fig. 3. The second column gives the simulated degree of complexity, the third column the AIDA campaign name and AIDA experiment number. The fourth, fifth and sixth columns give the experiment start conditions: the start temperature, the used ice nuclei (IN) and the number concentration of the aerosol acting as a cloud condensation nuclei (CCN).

| | Simulated mesoscopic scale complexity | Campaign and experiment number | Starting temperature (K) | IN | CCN [$cm^{-3}$] |
|---|---|---|---|---|---|
| AIDA hom. | severely complex | RICE03, 36 | 223 | Sulphuric acid | 105 |
| AIDA het. 30% | severely to medium complex | RICE03, 42 | 223 | Soot | 32 |
| AIDA het. 20% | medium complex to pristine | RICE03, 43 | 223 | Soot | 35 |
| AIDA het. 1% | pristine | RICE02, 08 | 223 | Soot | 52 |

**Table 2.** Overview of the measurement campaigns. Temperature range (minimum, maximum and mean) during measurements in ice containing clouds, the operated instrumentation, the number of ice particles included in the analysis and the percentage of ice particles rejected from the analysis owing to shattering. No particles were rejected from the SOCRATES dataset since only PHIPS datasets with manually classified images were included.

| Campaign | $T_{min}$ (K) | $T_{max}$ (K) | $T_{mean}$ (K) | Instruments | Number of ice particles analyzed | Percentage of ice particles rejected owing to shattering |
|---|---|---|---|---|---|---|
| ACRIDICON | 198 | 240 | 216 | SID-3 & PHIPS | 28,123 (SID-3) & 78,177 (PHIPS) | 1.33% (SID-3) & 29.6% (PHIPS) |
| MAXPEC | 205 | 240 | 227 | SID-3 | 24,769 | 0.80% |
| ML-CIRRUS | 207 | 241 | 222 | SID-3 | 9,830 | 0.07% |
| RACEPAC | 260 | 273 | 267 | SID-3 | 1,069 | 19% |
| ACLOUD | 256 | 281 | 271 | SID-3 & PHIPS | 2,812 (SID-3) & 20,610 (PHIPS) | 7.46% (SID-3) & 22.3% (PHIPS) |
| ARISTO 2017 | 215 | 259 | 239 | PHIPS | 9,984 | 15.2% |
| CIRRUS'98 | 218 | 233 | 230 | PN | 2,000 | |
| ASTAR | 265 | 271 | 268 | PN | 2,000 | |
| CONCERT | 213 | 258 | 227 | PN | 4,500 | |
| INCA NH | 208 | 240 | 227 | PN | 22,000 | |
| INCA SH | 213 | 240 | 227 | PN | 32,000 | |
| SOCRATES | 238 | 277 | 251 | PHIPS | 107,945 | - |