# Peer review of "Additional Global Climate Cooling by Clouds due to Ice Crystal Complexity"

_Atmospheric Chemistry and Physics, 2018_

## Referee Comment (RC1) · Anonymous Referee #1 · 28 Jun 2018

**General comments**

The paper presents a compilation of light scattering measurements obtained from a large number of aircraft campaigns, distributed globally, and they relate these measurements to ice crystal submicron complexity. This enables the authors to obtain estimates for the asymmetry parameter, a parameter of importance in NWP and climate modeling. They find from their analyses that the asymmetry parameter determination of 0.75 can be related to their complexity findings. This appears invariant with location and ice/cirrus formation, and the resulting scattering pattern results from the observed ice crystal complexity. As a consequence, this complexity expressed through the asymmetry parameter induces a not insubstantial-averaged further cooling effect not currently accounted for in climate models.

This is a largely well written paper, which links experimental results with theory and relates these measurements to ice crystal complexity and follows the theory through to an application in climate models. The paper provides nice results which deserve to be published, but the claim needs to be proven more rigorously with uncertainties attached to their estimates.

Major comments

1. The claim of the authors is that their measured PN angular scattering patterns are sufficient to determine the asymmetry parameter through some theoretical phase function that appears to fit through the data. This is not convincingly shown to be the case and appear to be eye fits at one single wavelength. There is no discussion in the text as to how the best fit to the measurements was statistically determined? Moreover, there are a number of extrapolations that could be used owing to the spread throughout the data, what uncertainty does this spread produce in the estimated asymmetry parameter values? There should be an uncertainty attached to their estimate of 0.75±? Once these uncertainties have been derived for the asymmetry parameter, the uncertainty in the SWCRE should be consequently determined.

2. The other wavelength of 0.804 um is only once shown, the same as Figure 5 should be shown but for 0.804 um using all models. Moreover, the eight-column aggregate shown at 0.804 um, is only just within the measured uncertainties at side scattering angles. This could be owing to the aspect ratio of the monomer columns not being sufficiently large and spaced out more than the compact model they show. The aspect ratio is also an important determinant of the asymmetry parameter as shown by Fu (2007), among others. It would be interesting to plot the approach of Fu (2007), to see if that treatment provides similar low values to those being estimated from the data.

3. The paper concludes that it is appropriate to apply the eight-column aggregate in climate and weather models. This is a rather significant claim as the model has only been tested at one single wavelength, at 0.805 um, it does not appear to possess the

correct absorption properties at side scattering angles for the possible reasons stated above. It is unclear as to how this model would fit observations at other wavelengths of importance, such as in the terrestrial window region, far infrared, and at more absorbing solar wavelengths, such as at 1.6 and 2.2 um. These wavelengths are also of importance in weather and climate modelling. The authors present no evidence to support their general claim.

4. A further point about Figure 5 also needs to be noted. Recent theoretical electromagnetic studies have shown that surface roughness, at scattering angles around exact backscatter, induces coherent backscattering, so the phase functions of surface roughened ice should not apparently be flat at exact backscattering angles, there ought to be some backscattering amplitude present. The authors are referred to the following paper for further information about this interesting interference effect, https://www.osapublishing.org/DirectPDFAccess/B8203150-AE8E-68E9-D2CB7062A1AB5EF8_385794/oe-26-10-A508.pdf?da=1&id=385794&seq=0&mobile=no . To compute the phase functions, the authors use a database which probably applies the improved physical optics approximation, in that multiple scattering is not included, so surface roughness is approximated by some geometrical treatment such as facet tilting to smooth the phase functions that appear in Figure 5. As a consequence of this, one could argue that the phase functions presented in Figure 5 are incorrect. Of course, owing to the asymmetry parameter being largely determined by diffraction, its derived value will not be much affected by this backscattering amplitude. However, this still does need to be noted in my opinion to encourage inclusion of multiple scattering in calculating the phase functions, especially if they are to be used for lidar applications at visible wavelengths. However, to obtain more representative phase functions, the backscattering amplitude could be added on to the phase functions presented in Figure 5. There is a parameterization that the authors could use to do this as explained in this paper  https://www.osapublishing.org/oe/abstract.cfm?uri=oe-24-1-620,  where  IGOM is corrected using the estimated amplitude obtained from electromagnetic calculations.

[Figure]

5. Also, for some reason, the authors do not cite papers prior to 2010, there are some, but these are few and far between and tend to be their own. This needs to be corrected.

Minor comments now follow:

1. In the abstract, the averaged asymmetry parameter of 0.75 is determined at the wavelength of?

2. Introduction line 15, similar results by Ulanowski et al., (2006) and Ulanowski et al. 2014 were also reported.

3. Introduction line 16, representations of ice crystal surface roughness via facet tilting were also added prior to 2008 by Macke et al. (1996)[ https://journals.ametsoc.org/doi/pdf/10.1175/1520-0469%281996%29053%3C2813%3ASSPOAI%3E2.0.CO%3B2], Yang and Liou (1998) [Single-scattering properties of complex ice crystals in terrestrial atmosphere, Contr. Atmos. Phys., 71, 223–248, 1998], Baran et al, (2001)[ https://rmets.onlinelibrary.wiley.com/doi/abs/10.1002/qj.49712757711], Baran and Francis (2004)[ https://rmets.onlinelibrary.wiley.com/doi/10.1256/qj.03.151], Sun et al. (2004)[ https://www.osapublishing.org/ao/abstract.cfm?uri=ao-43-9-1957. There are of course others.

9. Page 2, discussion on polarization, line 2, The same was also shown by Baran and Labonnote (2006) [https://www.sciencedirect.com/science/article/pii/S0022407305003699] in regards to polarization.

10. Page 3, line 15, replace "in" by "on".

11. Page 3, line 25, perhaps, the word "the" needs to be incorporated before "discrete dipole".

12. Page 3, line 34, insert the word "to" before "as"...

13. Section 2.2, in the discussion on the PN being used to determine the angular scattering functions, there is no explanation or discussion as to how shattered artefacts were removed from the analysis. Please could you insert this, otherwise, we may be led to believe that those functions could be more pertinent to shattered ice and so will provide low asymmetry parameter estimates.

14. Section 2.4, perhaps save space by compiling the list of campaigns into a table? This improves readability.

15. There are many campaigns dating back to before 2010, how did the authors make sure that the PSDs were treated consistently into one database from the variety of differing microphysical probes?

16. Page 5, line 23, suggest replace "to" with "for" . . . the analysis. . .

17. Section 2.5, please add a description of the current ice optical parameterization used in ECHAM-HAM. It is often referred to but unknown as to what it actually is.

18. Page 7, line 6, suggest insert the word "to". . . a change. . . .

19. Page 7, line 10, comma after aggregates?

20. Page 8, there are a whole list of studies that predate 2010 in showing that flat featureless phase functions best represent angular short-wave measurements obtained from above ice cloud such as Doutriaux-Boucher et al., (2000)[ https://agupubs.onlinelibrary.wiley.com/doi/abs/10.1029/1999GL010870 ], Labonnote et al. (2001)[ https://agupubs.onlinelibrary.wiley.com/doi/abs/10.1029/2000JD900642 ]. A more recent paper by Letu etal. (2016) [https://www.atmos-chem-phys.net/16/12287/2016/] uses comprehensive PARASOL short-wave reflectance data to show the same.

21. Page 8, line 16, Again, there are many papers that predate 2013, please cite a representative sample.

22. Page 9, line 5, typo "sdiscussed".

Figures:

Fig. 1 penale-> panel.

Fig. 2 difficult to distinguish purple from red, suggest changing purple to green.

Table 2. Please also insert the percentage of the total particle population rejected owing to shattering.

Please also note the supplement to this comment:
https://www.atmos-chem-phys-discuss.net/acp-2018-491/acp-2018-491-RC1-supplement.pdf

---

## Referee Comment (RC2) · Anonymous Referee #2 · 12 Jul 2018

Comments on 'Additional Global Climate Cooling by Clouds due to Ice Crystal Complexity'

By Järvinen et al.

Submitted to atmospheric chemistry and physics

In this article, ice particle complexity was derived from field campaigns spread over the globe, and it was further compared to chamber study. Angular light scattering functions from measurements were compared to Ping Yang's models, and it was concluded that roughened column aggregates model is the best representative of measurements. The new asymmetric factor derived from roughened column aggregates was explored in changing cloud radiative effects using a climate model. Overall, this article is well constructed and novel. Particularly, comparison of phase function between measurement and theoretical model will benefit other research areas such as model parameterization or remote sensing. My general comments: 1) explain how to obtain SWCRE from ECHAM model; 2) indicate how large biases of phase function exist between smooth and roughened particles.

Suggestion is to accept after a minor revision.

**Specific comments:**

Lines 15-17: ' reduce the SWCRE by 1-2 W m$^{-2}$' is confusing. It reads like that the magnitude of SWCRE is reduced, i.e. SW cooling is reduced by lowering g. This is conflicted with your conclusion. Please double check Yi et al. 2013 and make it clear.

Line 29: Please indicate what are 'two instruments'.

Figure 1" upper penale → upper panel ;  In lower panel, some scales are not clear.

Page 5 line 29: 'In these campaigns', do you mean all arctic campaigns ? Are there mixed phase clouds in SOCRATES campaign or midlatitude campaigns such as ARISTO 2017 and CONCERT with relatively high temperatures?

Page 6, section 2.5: Could you explain clearly how do you obtain SWCRE from ECHAM? Is it a parameter output from ECHAM, or do you run a radiative transfer model using ice clouds output from ECHAM?

Line 20: 'For generation of the theoretical phase functions…..', do you mean that the phase function here is not for only one particle, instead for integration of a series of particles like bulk property?

Figure 4 and Figure 5: are the measured 'angular light scattering functions' the same in both figures? If yes, please indicate. Also, roughened particles are used here for comparison because studies indicate that they perform well in many applications. How would the smooth particle model curve look like if they are overplotted in Figures 4 and 5?

Line 8: ' the global mean change in the SWCRE is -1.12 W m$^{-2}$', please indicate that more cooling is brought in using new g parameterization.

Line 9: ' the change in global SWCRE is small compared to ….', yes, it is right. However, based on Section 2.5, SWCRE is for ice clouds only. Is the change significant relative to your simulated SWCRE with new and old parameterizations? How about compare to SWCRE by ice clouds from [*Gasparini and Lohmann*, 2016] and [*Hong et al.*, 2016] where show ice cloud radiative effect using ECHAM-HAM model and from observations?

Line 11: ' the decrease in SWCRE…', please indicate cooling is enhanced.

'cirrus CRE', please explain what ice clouds have been used for CRE studied? Thin cirrus only?

References

Gasparini, B., and U. Lohmann (2016), Why cirrus cloud seeding cannot substantially cool the planet, *J. Geophys. Res. Atmos.*, 1–17, doi:10.1002/2015JD024666.

Hong, Y., G. Liu, and J.-L. F. Li (2016), Assessing the Radiative Effects of Global Ice Clouds Based on CloudSat and CALIPSO Measurements, *J. Clim.*, (2011), In press, doi:10.1175/JCLI-D-15-0799.1.

---

## Referee Comment (RC3) · Anonymous Referee #3 · 17 Jul 2018

Review of E. Järvinen et al., 2018

This paper describes the submicron scale complexity of individual ice crystals derived from airborne measurements and cloud chamber experiments. The authors assess that a new radiation parameterization for global climate models considering the higher roughness of ice crystals reveals a lower SWCRE. I find the paper very well written, logically organized, and the figures and tables are appropriate. I recommend the paper to be published with minor revision.

Special comments:

1. Page 2, Line 30/31: "In two cases the crystal complexity measurements and the angular light scattering measurements were conducted on the same ice particle pop-

ulation." I did not get where you use this coupled information later. Or is there any advantage at all having the measurements on the same ice particle population?

2. Page 3, Line 22/23: Why is there more shattering in mixed-phase clouds?

3. Page 8, Line 20: Why is the size distribution from the ACRIDICON-CHUVA campaign representive?

4. Page 10, Line 8: Why are these regional differences in the change of SWCRE? Why is the signal mainly in the tropics?

5. Page 11, Line 4: Before you could investigate the role in a warmer climate, you need to know if there are changes of the submicron scale complexity in a warmer climate. Do you expect them?

Technical corrections:

1. Page 3, Line 3; Page 5, Line 2; Page 7, Line 8: "sub-micron". Mostly you write "submicron", hyphenless.

---

## Author Comment (AC3) · 28 Sep 2018

**Authors' Response to Anonymous Referee #3 Comments**

The authors would like to thank the anonymous Referee for her/his helpful comments. Please find below a detailed point-by-point replies to each comment. Referee's comments are in blue and authors' replies in black.

This paper describes the submicron scale complexity of individual ice crystals derived from airborne measurements and cloud chamber experiments. The authors assess that a new radiation parameterization for global climate models considering the higher roughness of ice crystals reveals a lower SWCRE. I find the paper very well written, logically organized, and the figures and tables are appropriate. I recommend the paper to be published with minor revision.
We thank the anonymous Referee for this very encouraging general comment. Below we address the suggested minor revisions.

Special comments:
1. Page 2, Line 30/31: "In two cases the crystal complexity measurements and the angular light scattering measurements were conducted on the same ice particle population." I did not get where you use this coupled information later. Or is there any advantage at all having the measurements on the same ice particle population?
One of the main conclusions in this manuscript is that the high degree of ice crystal complexity is the reason why a similar angular scattering function is measured at different geographical locations. Two cases, where the two measurements ice crystal complexity and angular light scattering measurements) were performed on the same particle population, helps justify this conclusion.

2. Page 3, Line 22/23: Why is there more shattering in mixed-phase clouds?
Shattering is enhanced by the presence of large ice crystals and ice crystals with certain habits, such as bullet rosettes or large aggregates. In mixed-phase clouds, there tend to be more precipitation-sized ice crystals and rimed (aggregated) ice crystals. Jackson et al. (2014) showed that at temperatures > -8°C more shattering is present than in colder temperatures due to the presence of rimed ice crystals. These observations would explain the higher fraction of shattering in mixed-phase clouds since most of these measurements happened at temperatures warmer than -8°C.

We modified the sentence on page 3, line 22/23 the following: "*In mixed- phase clouds a higher fraction of measured 2-D scattering patterns, between 7.5% and 19%, were excluded from analysis. The higher fraction of shattering in mixed-phase clouds can be explained by the presence of rimed particles (Jackson et al., 2014)*".

3. Page 8, Line 20: Why is the size distribution from the ACRIDICON-CHUVA campaign representative?
The size distribution from the ACRIDICON-CHUVA campaign is measured for the same particles whose angular scattering functions are shown, i.e. the size distribution corresponds to the scattering information. It was also investigated how sensitive the retrieved angular scattering function is to the assumed size distribution and it was found to be insensitive to small changes in the median diameter. This is explained in the text at page 8 lines 22 and 23 (old manuscript version). Therefore, the exact size distribution is not crucial for finding the best fit.

4. Page 10, Line 8: Why are these regional differences in the change of SWCRE? Why is the signal mainly in the tropics?
The regional differences are linked with the cirrus occurrence. The cirrus occurrence is the highest in the tropics which, consequently, leads to the largest change in the SWCRE. We added a sentence in the chapter 4.2 to explain the regional differences: "*The largest effect is found in the tropical regions where also the cirrus occurrence is the highest (e.g. Sassen et al., 2008).*"

This is a difficult question to answer as long as we do not really understand the origin of submicron scale complexity in ice crystals. The only evidence we have is that abundance of heterogeneous ice nucleating particles (INPs) might decrease the supersaturation needed to nucleate the ice and, therefore, could lead to decrease in ice crystal mesoscopic complexity. However, increase in INPs also changes the cloud optical depth, which leads to another forcing.

With the statement on page 11 line 4 we wanted to motivate studies to investigate how cirrus radiative forcing would change if mesoscopic complexity is included in the projections versus if assuming the standard parameterisation.

We changed the term "sub-micron complexity" to "mesoscopic complexity" in the entire text. The reason for this was a critique towards presenting a new term to the field whereas the term "mesoscopic complexity" is already established.

References

Jackson, R. C., McFarquhar, G. M., Stith, J., Beals, M., Shaw, R. A., Jensen, J., Fugal, J., and Korolev, A.: An assessment of the impact of antishattering tips and artifact removal techniques on cloud ice size distributions measured by the 2D cloud probe, Journal of Atmospheric and Oceanic Technology, 31, 2567–2590, 2014.

---

## Author Comment (AC4) · 28 Sep 2018

**Authors' Response to Interactive Comment by Z Ulanowski**

The authors appreciate the Interactive Comments made by Z Ulanowski. Below we provide our answers to the raised issues. The Interactive Comments are in blue and authors' replies in black.

This extensive study investigates a very important area concerning the radiative impact of atmospheric ice. It could make an important contribution to this subject. However, several conclusions being made are too strong in my view and should be qualified. There is also one large flaw that should be addressed to increase the value of the study.

We thank Z Ulanowski for acknowledging the importance of this study and address his comments below.

4.2 p.10. My main point is a significant weakness of this study, the omission of long-wave (LW) effects of cirrus. To illustrate the importance of this shortcoming, the cirrus radiative effect difference found here is dominated by changes in the Tropical Warm Pool (TWP) and Maritime Continent. Yet in this region the net radiative influence of cirrus is determined largely by the longwave, with difference from even the zonal average of the order of many tens of W/m2 (e.g. Xu and Guan, 2017; NOAA/ESRL), in contrast to the \_peak\_ SW value of about 8W/m2 reported here. So potentially not just the magnitude but even the sign of the postulated effect could change. Hence the LW effect should be taken into account. The severely roughened hexagonal aggregate model that is adopted by the authors includes IR properties. Why were they not included to obtain the net radiative effect? Was the longwave parameterization done but the effects are not shown - why, it should be easy to do? Or was the parameterization not applied - which makes the model internally inconsistent? If this result is being kept "for later", I would strongly advise against it - salami-slicing climate science is a risky undertaking, e.g. the longwave cloud feedback is reported to be positive, mostly due to tropical cirrus (Zelinka and Hartmann, 2010), potentially negating the main conclusion from the work.

In this study we only discuss the effect of ice crystal complexity to the SWCRE and omitting the LW effect will not in any way change the conclusion of this work. There are two reasons why we do not discuss the LW effect. First, the focus of this study is the effect of ice crystal complexity on the ice cloud asymmetry factor. In the ECHAM-HAM model the ice particle asymmetry factors are only considered for calculation of the SW effect and are not included in the calculations of the LW effect. This is due to the fact that the LW effect is less sensitive to the ice crystal morphology than the SW effect. For example, Yi et al. (2013) showed that changing the ice crystals from smooth to complex will not significantly affect the LWCRE.

Secondly, our optical measurements are in the SW region and, therefore, we can only make conclusion of the SW asymmetry factors. We agree that optical measurements in the LW region would be of interest to validate LW parameterizations in the future. Furthermore, we think that the term "salami-slicing" is more than misplaced with regard to this work, which - in our opinion - represents one of the most comprehensive studies on ice crystal complexity and its influence on the cloud radiative forcing. The experimental data used here are from dedicated cloud chamber simulation runs as well as from the field, gathered in a dozen of aircraft projects around the globe. Further, the data are used to construct a new, more realistic parameterization of the asymmetry factor to be used in climate models - a scientific span that is not common in the field.

This brings me to a related point: the authors make strong statements about the radiative impact, with the largest impact being demonstrated in the TWP/MC region. Yet no in situ data from this region is provided, and very little data from the tropics altogether. What there is, refers to Amazonia, where modelling indicates very weak impact.

The data presented in this study covers all the geographical areas where the KIT SID-3 and the PHIPS instruments have been flown and a large amount of the campaigns where PN measurements were available. We agree that the TWP/MC region (where these instruments have not yet flown) is highly important for the cirrus cloud radiative impact. We hope that in future more field campaigns will be focused on this area, where this study demonstrates the largest SW impact.

Some smaller points follow.

Introduction p.2 and section 2.1 p.3. I find it surprising that the authors do not properly acknowledge that SID3, the core instrument in this work, and long-term assistance with the hardware, software and data analysis techniques were provided to KIT by the team at University of Hertfordshire.

The SID-3 instrument was developed by the University of Hertfordshire and a version of the instrument was purchased by KIT in 2008. It is true that many collaborative efforts between KIT and Hertfordshire has taken place to improve the hardware, software and data analysis methods to the current status and we believe that these collaborative efforts are correctly documented in the corresponding literature. The instrument itself is cited through the original Hertfordshire publication of Kaye et al. (2008). The University of Hertfordshire was involved in the first field deployment of the instrument in the MACPEX campaign, which is acknowledged by co-authorship in the Järvinen et al. (2016) and Schmitt et al. (2016b) publications. The SID-3 scattering pattern analysis methods for atmospheric ice particles were also developed in close collaboration between KIT an University of Hertfordshire by conducting at least five joint AIDA cloud simulation campaigns. This effort is acknowledged in the original work describing the use of the complexity parameter,  $k_e$ , as a complexity measure (Schnaiter et al., 2016), where the University of Hertfordshire et al., 2016) is listed as co-author.

**2.1 p.3. Likewise, the method for determining ice crystal roughness using pattern texture analysis (including GLCM) was developed by the Hertfordshire group (Ulanowski et al., 2010, 2014). This should be acknowledged too.**

Analysing scattering patterns to retrieve information on surface roughness has been previously used in industrial applications for surface quality control (e.g. Lu et al., 2006) but it is true that the Hertfordshire group was the first to use this technique for ice crystal surface roughness. Therefore, we have added the citation to Ulanowski et al., 2010, 2014 to the following sentence: *"The crystal complexity is quantified from the 2-D scattering patterns using a grey-level co-occurrence matrix (GLCM) method (Lu et al., 2006). This method was developed for industrial quality control of surface treatment processes but was later adapted for analysis of complexity features of three-dimensional ice particles (Ulanowski et al., 2010, 2014; Schnaiter et al., 2016)."*

3.2 p.7. "enhanced submicron scale complexity of homogeneously formed ice crystals [...] and can be explained by an increased stacking disorder of homogeneously nucleated ice crystals" Firstly, it would be difficult to associate in situ measurements with the homogeneous mode of nucleation in such categorical fashion. The second part of this statement is extremely simplistic too, no proof of a general connection of complexity with stacking disorder exists yet, even in the lab let alone the atmosphere. While stacking-disordered ice has been produced in the supercooled water freezing experiments of Malkin et al. (2012), heterogeneous ice nucleation is equally important and there can be other reasons why roughness arises (Chou et al., 2018). We refer to the in situ measurements that were presented in Ulanowski et al. (2014). The authors argued that in situ observations in a mid-latitudes cirrus showed differences in the ice crystal complexity based on the airmass origin: "*polluted airflow showed significantly lower roughness for all measures apart from kurtosis. We speculate that this was due to higher concentration of inhomogeneous ice nuclei (IN) in the last case"*. Of course it is difficult to investigate the origin of ice crystal complexity based on in situ measurements, especially if the ice particle history is unknown. Therefore, such laboratory studies will be valuable to interpreted in situ field results.

For the second point, we agree that our knowledge of formation of surface roughness in a single crystal is still highly unknown. Therefore, we modified the sentence as: "*can be partly explained by...*".

3.2. p7. While cyclic growth has been shown to contribute to increased ice roughness (Chou et al., 2018) the SEM experiments that are cited (Magee et al., 2014) are thought to have limited relevance to ice behaviour at tropospheric conditions, as growth in the near-vacuum of a SEM takes place under kinetically-limited, not diffusion-limited conditions typical of the troposphere (Kiselev, 2014; Chou et al., 2018).

We agree that discussing the results of SEM experiments in atmospheric context is challenging due to the near-vacuum pressure conditions experienced by the ice crystals. Therefore, it is important to have proof such results in atmospheric conditions as shown in Chou et al. (2018). We have added this reference to the sentence together with the Magee et al. (2014) reference.

**References**

Chou C., Voigtländer J., Ulanowski Z., Herenz P., Bieligk H., Clauss T., Niedermeier D., Hartmann S., Ritter G., Stratmann F.: Ice crystals roughness during depositional growth and sublimation, Atm. Chem. Phys., doi:10.5194/acp-2018-254, in review, 2018.

Kiselev, A.: Interactive comment on "Mesoscopic surface roughness of ice crystals pervasive across a wide range of ice crystal conditions" by N. B. Magee et al., Atmos. Chem. Phys. Discuss., 14, C4758–C4763, http://www.atmos-chem-phys- discuss.net/14/C4758/2014/, 2014. Malkin, T. L., Murray, B. J., Brukhno, A. V., Anwar, J., and Salzmann, C. G.: Structure of ice crystallized from supercooled water, Proceedings of the National Academy of Sciences, 109, 1041–1045, 2012.

NOAA/ESRL http://www.esrl.noaa.gov/psd/

Ulanowski Z., P.H. Kaye, E. Hirst & R.S. Greenaway: Light scattering by ice particles in the Earth's atmosphere and related laboratory measurements, In: Proc. 12th Int. Conf. Electromagnetic & Light Scatt., Helsinki, 294-297, 2010.

Zelinka, M. D., and D. L. Hartmann: Why is longwave cloud feedback positive?, J. Geophys. Res., 115, D16117, doi: 10.1029/2010JD013817, 2010.

Xu, Q. and Guan, Z.: Interannual variability of summertime outgoing longwave radiation over the Maritime Continent in relation to East Asian summer monsoon anomalies, J. Meteorological Research, 31, 665-677, 2017.

**References**

Järvinen, E., Schnaiter, M., Mioche, G., Jourdan, O., Shcherbakov, V. N., Costa, A., Afchine, A., Krämer, M., Heidelberg, F., Jurkat, T., Voigt, C., Schlager, H., Nichman, L., Gallagher, M., Hirst, E., Schmitt, C., Bansemer, A., Heymsfield, A., Lawson, P., Tricoli, U., Pfeilsticker, K., Vochezer, P., Möhler, O., and Leisner, T.: Quasi-Spherical Ice in Convective Clouds, Journal of the Atmospheric Sciences, 73, 3885–3910, https://doi.org/10.1175/JAS-D-15-0365.1, 2016.

Kaye, P. H., Hirst, E., Greenaway, R. S., Ulanowski, Z., Hesse, E., DeMott, P. J., Saunders, C., and Connolly, P.: Classifying atmospheric ice crystals by spatial light scattering, Opt. Lett., 33, 1545–1547, 2008.

Lu, R.-S., Tian, G.-Y., Gledhill, D., and Ward, S.: Grinding surface roughness measurement based on the co-occurrence matrix of speckle pattern texture, Appl. Opt., 45, 8839–8847, 2006.

Schmitt, C. G., Schnaiter, M., Heymsfield, A. J., Yang, P., Hirst, E., and Bansemer, A.: The microphysical properties of small ice particles measured by the Small Ice Detector-3 probe during the MACPEX field campaign, Journal of the Atmospheric Sciences, 2016b.

---

## Author Response (AR1)

**Authors' Response to Anonymous Referee #1 Comments**

The authors would like to thank the anonymous Referee for her/his comments that have helped us to improve this manuscript. Below, the major and minor comments are addressed by detailed point-by-point replies. Referee's comments are in blue and authors' replies in black.

General comments

The paper presents a compilation of light scattering measurements obtained from a large number of aircraft campaigns, distributed globally, and they relate these measurements to ice crystal submicron complexity. This enables the authors to obtain estimates for the asymmetry parameter, a parameter of importance in NWP and climate modeling. They find from their analyses that the asymmetry parameter determination of 0.75 can be related to their complexity findings. This appears invariant with location and ice/cirrus formation, and the resulting scattering pattern results from the observed ice crystal complexity. As a consequence, this complexity expressed through the asymmetry parameter induces a not insubstantial-averaged further cooling effect not currently accounted for in climate models.

This is a largely well written paper, which links experimental results with theory and relates these measurements to ice crystal complexity and follows the theory through to an application in climate models. The paper provides nice results which deserve to be published, but the claim needs to be proven more rigorously with uncertainties attached to their estimates.

We thank the Referee for this positive general comment. In her/his comments the Referee has raised concerns on the rigour of the presented conclusions, especially on the uncertainties in the asymmetry factor. We acknowledge that the limitations of our measurements were not adequately discussed and have modified the discussion and conclusions to be more sensitive to these limitations. Below are listed the detailed replies to the Referee's major and minor comments.

Major comments

1. The claim of the authors is that their measured PN angular scattering patterns are sufficient to determine the asymmetry parameter through some theoretical phase function that appears to fit through the data. This is not convincingly shown to be the case and appear to be eye fits at one single wavelength. There is no discussion in the text as to how the best fit to the measurements was statistically determined? Moreover, there are a number of extrapolations that could be used owing to the spread throughout the data, what uncertainty does this spread produce in the estimated asymmetry parameter values? There should be an uncertainty attached to their estimate of 0.75±? Once these uncertainties have been derived for the asymmetry parameter, the uncertainty in the SWCRE should be consequently determined.

Estimating asymmetry factor from measurements that are covering only part of the angular region is challenging, especially since majority of the scattered intensity is found in the forward direction that is not covered by the measurements. We think that the best approach to estimate the asymmetry factor from the measurements is to use a physical model with a known asymmetry factor to fit the measurements in the known angular range, as done in this manuscript.

The same difficulty applies to estimating the uncertainty of the asymmetry factor if only part of the angular range is covered. For example, if we calculate the partial asymmetry factor for the angular range of 18 to 170° using the column aggregate model, we get an partial asymmetry factor of 0.14. This is approximately 20% of the total asymmetry factor of 0.75, since the forward peak contributes the majority part of the asymmetry factor. Even if we can estimate the uncertainty between the fit and the measurements in the restricted angular range, this uncertainty estimation contributes only 20% to the total uncertainty. Therefore, the measurements alone are not enough to estimate the uncertainty in the asymmetry factor, although it can be argued that the uncertainty of the asymmetry factor is the highest in the measurement region compared to the forward region, where the scattering intensity is mainly determined by the particle size and less of shape.

Owing to this discussion, we agree with the Referee that the discussions in the Sect. 3.4 and in the abstract and some of the conclusions are not well justified. Better than *retrieving* the

asymmetry factor from the measurements it is more justifiable to *compare* the different optical models to the measurements. Therefore, we have reformulated the Sect. 3.4 and the section title as "Comparison of the measured angular scattering functions to a light scattering database". We have also omitted the sentence "*Using the severely roughened hexagonal aggregate model asymmetry factors of 0.750 and 0.754 at 532 nm and 804 nm, respectively, were retrieved*" and instead write in the Sect. 4: "*the severely roughened hexagonal aggregate model has relatively low asymmetry factors of 0.750 and 0.754 for 532 nm and 804 nm, respectively*". The abstract we have modified so that instead of writing: "*as a consequence, a low asymmetry factor of 0.75 is observed*", we write "*as a consequence, a similar flat and featureless angular scattering function is observed. A comparison between the measurements and a database of optical particle properties showed that severely roughened hexagonal aggregates optimally represents the measurements in the observed angular range*".

We also make a stronger case in the Sect. 3.4 that the severely roughened hexagonal aggregate model best represents our measurement. We have added a new figure that compares the different models with the measurements at 804 nm (please see answer to comment 2) and justify the fit by calculating the root mean square errors (RMSE) between the model and the mean of the measurements. For both wavelengths the column aggregate model has the lowest RMSEs of 0.0017 and 0.0014 (for 532 and 804 nm, respectively) compared to the other models (RMSE between 0.0022 and 0.0111 for 532 nm, and 0.0037 and 0.0208 for 804 nm). The discussion of the RMSE analysis was added to the Sect. 3.4.

The asymmetry factors around 0.75 are, therefore, not *retrieved* from the measurements but represent the asymmetry factor of the severely roughened hexagonal aggregate model. This asymmetry factor is fixed for a given size distribution. Later, we use the severely roughened hexagonal aggregate model for deriving the parameterization of SW asymmetry factors for the ECHAM-HAM model. We believe this approach is still justified based on the microphysical observations. However, we agree that the claim in the last sentence of the first conclusions paragraph is not well justified and this conclusion is rewritten in the revised version (please see the reply to comment 3).

2. The other wavelength of 0.804 um is only once shown, the same as Figure 5 should be shown but for 0.804 um using all models. Moreover, the eight-column aggregate shown at 0.804 um, is only just within the measured uncertainties at side scattering angles. This could be owing to the aspect ratio of the monomer columns not being sufficiently large and spaced out more than the compact model they show. The aspect ratio is also an important determinant of the asymmetry parameter as shown by Fu (2007), among others. It would be interesting to plot the approach of Fu (2007), to see if that treatment provides similar low values to those being estimated from the data.

We added a new figure (Fig. 6) showing a comparison between the PN measurements and different models results calculated for 804 nm. This comparison shows that from all of the different habit models, the severely roughened column aggregate model has the best overall agreement with the measurements at 804 nm. Other models underestimate the backscattering intensity from 120° onwards. A discussion of this comparison was added in Section 3.4.

The Referee also suggests to modify the aspect ratio of the severely roughened column aggregate model in order to find a better fit to the measurements at the 804 nm wavelength, as done in the work of Fu (2007). Fu (2007) has showed that modifying the aspect ratio will influence the asymmetry factor of single hexagonal columns. We agree that modifying the aspect ratio of the severely roughened column aggregate model to create a better fit at 804 nm would be an interesting investigation. However, modifying the existing optical particle model or introducing new optical models is not the scope of this paper for two reasons. First, the focus of this paper is to present globally distributed observations of ice crystal mesoscopic complexity and relate them to the angular light scattering measurements. The modelling efforts in this paper are used as a tool to understand the implications of the measurement results. We hope that the measurements will inspire to development of new optical particle models in the future. Secondly, we do not have enough spectral information on the angular scattering functions that we can justify using different optical models for different wavelength bands. We think that the best approach here is to use one optical model to calculate the asymmetry factors for all the SW bands. The use of the severely

roughened column aggregate model is justified since it provides the best overall fit on both wavelengths.

3. The paper concludes that it is appropriate to apply the eight-column aggregate in climate and weather models. This is a rather significant claim as the model has only been tested at one single wavelength, at 0.805 um, it does not appear to possess the correct absorption properties at side scattering angles for the possible reasons stated above. It is unclear as to how this model would fit observations at other wavelengths of importance, such as in the terrestrial window region, far infrared, and at more absorbing solar wavelengths, such as at 1.6 and 2.2 um. These wavelengths are also of importance in weather and climate modelling. The authors present no evidence to support their general claim.
We agree that the spectral consistency of optical particle models is one of the biggest challenges in current climate models and in remote sensing retrievals. Modelling the spectral dependency of the asymmetry factor is difficult, since atmospheric measurements are available in only few wavelengths. Additional challenge is posed through the fact that each of the operated polar nephelometer work at a single wavelength, and therefore, combining the polar nephelometric measurements to gain spectral information will inevitably contain uncertainties merging from different measurement setups. For example, in our case we cannot completely distinguish, which proportion of the difference we see between the measurements and the model are real and which is contributed by the measurement setup, different calibration procedures, etc.

We agree that the spectral uncertainty of the asymmetry factors is not adequately discussed in the text. To correct this, we modified the section 4.1 and added the following discussion at the beginning of the section:

*"Fig. 4 showed that the observed high degree of mesoscopic scale complexity dominates the angular scattering function over the ice crystal shape and a uniform angular scattering function is observed at two wavelengths (532 and 804 nm). Therefore, it is justified to use a single-habit optical ice particle model assuming severely roughened surfaces to compute the bulk optical properties of ice clouds. It was found that the severely roughened column aggregate model showed the best fit of the atmospheric measurements performed at both wavelengths. At 804 nm the model disagreed slightly with the measurements at the sideward angles (Fig. 4). This disagreement indicates that either the severely roughened column aggregate model does not accurately represent the spectral dependence of the asymmetry factors, or could also be related to systematic measurement uncertainties caused by using different measurement systems. However, since we only have information on the ice particle angular scattering properties at two wavelengths at the moment, only one optical particle model is used to parameterize the asymmetry factors."*

We also changed the last sentence in the first paragraph of the conclusions: *"Moreover, since the ice particle angular scattering functions did not vary significantly between different geographical locations, the modelling efforts of ice particle optical properties in future weather forecast and climate models will be simplified."*

4. A further point about Figure 5 also needs to be noted. Recent theoretical electromagnetic studies have shown that surface roughness, at scattering angles around exact backscatter, induces coherent backscattering, so the phase functions of surface roughened ice should not apparently be flat at exact backscattering angles, there ought to be some backscattering amplitude present. The authors are referred to the following paper for further information about this interesting interference effect, https://www.osapublishing.org/DirectPDFAccess/B8203150-AE8E-68E9- D2CB7062A1AB5EF8_385794/oe-26-10-A508.pdf?da=1&id=385794&seq=0&mobile=no . To compute the phase functions, the authors use a database which probably applies the improved physical optics approximation, in that multiple scattering is not included, so surface roughness is approximated by some geometrical treatment such as facet tilting to smooth the phase functions that appear in Figure 5. As a consequence of this, one could argue that the phase functions presented in Figure 5 are incorrect. Of course, owing to the asymmetry parameter being largely determined by diffraction, its derived value will not be much affected by this backscattering amplitude. However, this still does need to be noted in my opinion to encourage inclusion of multiple scattering in calculating the phase functions,

especially if they are to be used for lidar applications at visible wavelengths. However, to obtain more representative phase functions, the backscattering amplitude could be added on to the phase functions presented in Figure 5. There is a parameterization that the authors could use to do this as explained in this paper https://www.osapublishing.org/oe/abstract.cfm?uri=oe-24-1-620, where IGOM is corrected using the estimated amplitude obtained from electromagnetic calculations.

The Referee points out that recent theoretical electromagnetic studies have shown that surface roughness can lead to coherent backscattering enhancement at angles around exact backscattering whereas the theoretical functions of severely roughened particles showed in Figs. 5 and 6 do not take into consideration this effect. However, for the aspect of energy redistribution in the scattering process the backscattering enhancement has a negligible effect - as also stated by the Referee and, thus the derived asymmetry factor will not be affected through exclusion of this effect. The Referee also pointed out that this effect can have consequences for lidar application, which we agree. Therefore, we added a discussion of this effect to the Chapter 3.4:

*"At the angles around exact-backscattering the severely roughened column aggregate model predicts a relatively flat behaviour. However, recent modelling studies have indicated that the scattering intensities around exact backscattering angles should be enhanced due to coherent scattering (e.g. Zhou, 2018). Although this effect can be important for lidar applications, it does not significantly affect the redistribution of the energy in the scattering process and, thus, the magnitude of the asymmetry factor."*

Furthermore, the Referee states that it is arguable that the theoretical phase functions in Figs. 5 and 6 are incorrect due to the treatment of the surface roughness in the model by using the tilted-facet (TF) method. Although the TF method may not accurately represent the physical surface roughness, it has been shown that the TF method can be used to model the phase matrix element P11 with high accuracy (Liu, Panetta and Yang, 2013). Also, arguing whether an optical particle model is incorrect is usually based on comparison of models with more sophisticated models, rarely on a comparison with measurements. This study presents an comparison of one optical data base with atmospheric measurements. In future, it is certainly of interest to perform more such comparisons with also other optical particle models to address the question of which optical models perform the best for different applications.

5. Also, for some reason, the authors do not cite papers prior to 2010, there are some, but these are few and far between and tend to be their own. This needs to be corrected.
We have expanded the list of cited papers. Please refer to answers to minor comments 2, 3, 9, 20 and 21.

Minor comments now follow:

1. In the abstract, the averaged asymmetry parameter of 0.75 is determined at the wavelength of?
We added wavelength after the asymmetry factor.

2. Introduction line 15, similar results by Ulanowski et al., (2006) and Ulanowski et al. 2014 were also reported.
We added citation to Ulanowski et al. (2016) and Ulanowski et al. 2014 to line 15.

3. Introduction line 16, representations of ice crystal surface rough- ness via facet tilting were also added prior to 2008 by Macke et al. (1996)[ https://journals.ametsoc.org/doi/pdf/10.1175/1520-0469%281996%29053%3C2813%3ASSPOAI%3E2.0.CO%3B2], Yang and Liou (1998) [Single-scattering properties of complex ice crystals in terrestrial at- mosphere, Contr. Atmos. Phys., 71, 223–248, 1998], Baran et al, (2001)[ https://rmets. onlinelibrary.wiley.com/doi/abs/10.1002/qj. 49712757711], Baran and Francis (2004)[ https://rmets.onlinelibrary.wiley.com/doi/10.1256/qj. 03.151], Sun et al. (2004)[ https://www.osapublishing.org/ao/abstract.cfm?uri=ao-43-9-1957. There are of course others.
We added both the citations to Make et al. (1996), Yang and Liou (1998), Baran et al. (2001), Baran and Francis (2004) and Sun et al. (2004) and "e.g." before the citations.

9. Page 2, discussion on polarization, line 2, The

same was also shown by Baran and Labonnote [https://www.sciencedirect.com/science/article/pii/S0022407305003699] in regards to polarization.
We added this reference.

10. Page 3, line 15, replace "in" by "on".
We corrected this.

11. Page 3, line 25, perhaps, the word "the" needs to be incorporated before "discrete dipole".
We added "the".

12. Page 3, line 34, insert the word "to" before "as". . .
We corrected this.

13. Section 2.2, in the discussion on the PN being used to determine the angular scattering functions, there is no explanation or discussion as to how shattered artefacts were removed from the analysis. Please could you insert this, otherwise, we may be led to believe that those functions could be more pertinent to shattered ice and so will provide low asymmetry parameter estimates.
The Referee is correct that the influence of the shattering artefacts to the PHIPS and PN measurements are not adequately discussed in Section 2.2. Since PHIPS performs particle-by-particle measurements, it is possible to detect shattering events by investigating the particle inter-arrival times. The analysis of the particle inter-arrival times revealed two modes - one mode of short inter arrival times corresponding to shattering events and one mode of longer inter arrival times corresponding to real particle events. These two modes can be separated with a threshold of approximately 1 ms. We have removed all the angular scattering functions identified as shattered events from the analysis.

The analysis of the PN data, including shattering artefacts, is discussed in the original publications cited in this manuscript and in previous studies. For example, the effects of shattering artefacts to the PN measurements are discussed in the Appendix B of Mioche et al. (2017). The authors stated that although it is not possible to avoid or estimate the shattering effects in the PN signal, it can be estimated that the shattering artefacts are within the measurement uncertainty of the PN (25 % on the extinction coefficient).

Two sentences discussing the shattering effects were added to the Section 2.2 after discussion of both instruments:

PHIPS: *"Before analysis, particles corresponding to shattering events were removed by calculating particle inter-arrival times and removing particle pairs with inter-arrival times <1ms."*

PN: *"It is not possible to correct the PN data for shattering artefacts but it has been estimated that possible shattering artefacts contribute less than 25% to the total extinction signal (Mioche et al., 2017)."*

14. Section 2.4, perhaps save space by compiling the list of campaigns into a table? This improves readability.
Section 2.4 does not only give a list of the campaigns but also discusses the definition "ice cloud" in each of the campaigns (i.e. which cloud types were included in the analysis). The discussion presented in Sect. 2.4 is relevant for understanding the results, since different cloud systems were sampled in different campaigns. Furthermore, this section gives a description, how droplets were excluded from the dataset. For these reasons, we think Sect. 2.4 is important and cannot be reduced as a table.

15. There are many campaigns dating back to before 2010, how did the authors make sure that the PSDs were treated consistently into one database from the variety of differing microphysical probes?
This manuscript reports measurements from three different microphysical probes: the Small Ice Detector 3 (SID-3), the Particle Habit Imaging and Polar Scattering (PHIPS) probe and the Polar Nephelometer (PN). All the SID-3 and PHIPS data from each of the field campaigns are analyzed using the procedures described in the Sections 2.1 and 2.2. Also, the analysis methods of the PN probe have not been modified since published in Gayet et al. 1997.

16. Page 5, line 23, suggest replace "to" with "for" . . . the analysis. . .
We corrected this.

17. Section 2.5, please add a description of the current ice optical parameterization
used in ECHAM-HAM. It is often referred to but unknown as to what it actually is.
The current ice optical parameterization in the ECHAM-HAM is calculated with Mie-theory but the
asymmetry parameters are scaled down to be more reliable for aspherical ice particles. We added
this description of the current optical parameterization to Sect. 4, where the ECHAM-HAM model
is discussed.

18. Page 7, line 6, suggest insert the word "to". . . a change. . ..
We corrected this.

19. Page 7, line 10, comma after aggregates?
We added a comma.

20. Page 8, there are a whole list of studies that predate 2010 in showing that flat featureless
phase functions best represent angular short-wave measurements obtained from above ice cloud
such as Doutriaux-Boucher et al., (2000)[ https://agupubs.onlinelibrary.wiley.com/doi/abs/
10.1029/1999GL010870 ], Labonnote et al. (2001)[ https://agupubs.onlinelibrary.wiley.com/doi/
abs/10.1029/2000JD900642 ]. A more recent paper by Letu et al. (2016) [https://www.atmos-
chem- phys.net/16/12287/2016/] uses comprehensive PARASOL short-wave reflectance data to
show the same.
We extended the references as suggested.

21. Page 8, line 16, Again, there are many papers that predate 2013, please cite a representative
sample.
We extended the references by citing Macke et al. (1996), Yang and Liou (1998), Liou et al. (2000),
Baum et al. (2010), Baum et al. (2011), Baran (2012), Diedenhoven et al. (2012). We also added
e.g. before the references to illustrate that the cited references are a subsample of literature.

22. Page 9, line 5, typo "sdiscussed".
We corrected this.

Figures:
Fig. 1 penale-> panel.
We corrected this.

Fig. 2 difficult to distinguish purple from red, suggest changing purple to green.
We changed the purse trajectories to green and modified the colours in Fig. 3 accordingly.

Table 2. Please also insert the percentage of the total particle population rejected owing to
shattering.
We have added a new column to the table showing the percentage of ice particles rejected owing
to shattering.

**References**

Gayet, J.-F., Crépel, O., Fournol, J., and Oshchepkov, S.: A new airborne polar Nephelometer for
the measurements of optical and micro- physical cloud properties. Part I: Theoretical design,
Annales Geophysicae, 15, 451–459, 1997.

Liu C., R. L., Panetta, P., Yang, 2013: The effects of surface roughness on the scattering
properties with sizes from the Rayleigh to the geometric-optics regimes. *J. Quant. Spectrosc.
Radiat. Transfer.*, 129:169-185.

Mioche, Guillaume, et al. "Vertical distribution of microphysical properties of Arctic springtime low-level mixed-phase clouds over the Greenland and Norwegian seas." *Atmospheric Chemistry and Physics* 17 (2017): 12845-12869.

**Authors' Response to Anonymous Referee #2 Comments**

The authors would like to thank the anonymous Referee for her/his helpful comments. Please find below a detailed point-by-point replies to each comment. Referee's comments are in blue and authors' replies in black.

In this article, ice particle complexity was derived from field campaigns spread over the globe, and it was further compared to chamber study. Angular light scattering functions from measurements were compared to Ping Yang's models, and it was concluded that roughened column aggregates model is the best representative of measurements. The new asymmetric factor derived from roughened column aggregates was explored in changing cloud radiative effects using a climate model. Overall, this article is well constructed and novel. Particularly, comparison of phase function between measurement and theoretical model will benefit other research areas such as model parameterization or remote sensing. My general comments: 1) explain how to obtain SWCRE from ECHAM model; 2) indicate how large biases of phase function exist between smooth and roughened particles.

We thank the Referee for her/his encouraging comments. To address the general comment (1) we have added a detailed description on how SWCRE was obtained using the ECHAM model. Please see the answer to the specific comments below for more details.

In the general comment (2) the Referee asks to address the differences between the phase functions and asymmetry factors of smooth and severely roughened particles. For the database used in this manuscript, this difference is discussed in the original study of Yang et al. (2013). In Fig. 13 of their paper a comparison is shown of phase functions and asymmetry factors for smooth and roughened particles. The phase functions of smooth particles show minima and maxima peaks at certain angles whereas as the roughness is increased, these peaks are more smoothened out (please also see the answer to the specific comment below). Also, the intensity at the sideward angles is the highest for the roughest particles, which leads to lowering of the asymmetry factor as seen in the lower panels of Fig. 13.

These biases between the phase functions of smooth and roughened particles are discussed in several occasions in this manuscript (please see answers below) and in the cited research (e.g. in Yi et al. (2013)). We think that the given discussion together with the references give the reader a good overview of the effects of roughness on the phase function and on the asymmetry factor.

Suggestion is to accept after a minor revision.

**Specific comments:**

Lines 15-17: ' reduce the SWCRE by 1-2 W m-2' is confusing. It reads like that the magnitude of SWCRE is reduced, i.e. SW cooling is reduced by lowering g. This is conflicted with your conclusion. Please double check Yi et al. 2013 and make it clear.

We agree that the wording online 15-17 is confusing, so we reformatted the sentence as: "*Yi et al. (2013) showed that by assuming severely roughened ice crystals and, thus, lowering the cloud short wave (SW) asymmetry factors between 0.01 and 0.035, can cause additional SW cooling by 1-2 W m$^{-2}$*".

Line 29: Please indicate what are 'two instruments'.

We added the names of the two instruments: *"The observations of the ice crystal mesoscopic complexity are linked with measurements of the ice particle angular light scattering function at various geographical locations in the southern and northern hemisphere performed with two polar nephelometers, the Particle Habit Imaging and Polar Scattering (PHIPS) probe and the Polar Nephelometer (PN)."*

Figure 1" upper penale->upper panel ; In lower panel, some scales are not clear.
We corrected this and increased the font of the scales in the lower panel.

Page 5 line 29: 'In these campaigns', do you mean all arctic campaigns ? Are there mixed phase clouds in SOCRATES campaign or midlatitude campaigns such as ARISTO 2017 and CONCERT with relatively high temperatures?
Yes, we mean all arctic campaigns. We changed in the text *"In these campaigns"* to *"In the arctic campaigns"*.

We measured mixed-phase clouds in the SOCRATES campaign but in the midlatitude campaigns only measurements in fully glaciated clouds are included. To make this more clear we added the term "mixed-phase" on line 1 (page 6).

Page 6, section 2.5: Could you explain clearly how do you obtain SWCRE from ECHAM? Is it a parameter output from ECHAM, or do you run a radiative transfer model using ice clouds output from ECHAM?
We have added a detailed description of how SWCRE was obtained from the ECHAM model in the chapter 2.5.

*"The ECHAM-HAM model is used to calculate the SWCRE of the ice clouds, which is computed online by calling the radiation subroutine twice. The first call is with clouds (all-sky) and the second call is without clouds (clear-sky) in the atmosphere. The first call uses the standard model parameterization for the short wave asymmetry factors of ice clouds. The radiative fluxes from this call to the radiation subroutine are used to advance the model simulations. The cloud radiative effects are computed as the difference between the all-sky minus the clear-sky fluxes. To estimate the change in SWCRE by changing the short wave asymmetry factors of ice clouds an additional (third) call to the radiation subroutine is conducted. The additional (diagnostic) call to the radiation subroutine is identical to the first call except for using the new parameterization for the short wave asymmetry factors of ice clouds. The radiative fluxes from this additional call are only diagnostic. The SWCRE using the new parameterization for the short wave asymmetry factors of ice clouds is computed from the difference in SW radiative flux at the top of the atmosphere from the additional call and the cloud-free SW radiative flux at the top of the atmosphere."*

Line 20: 'For generation of the theoretical phase functions.....', do you mean that the phase function here is not for only one particle, instead for integration of a series of particles like bulk property?
The theoretical phase functions are calculated for orientation averaged particles that are integrated over a size distribution - similar to the measurements that are also averaged over a particle population. To make this more clear we modified the caption of Fig. 5: *"A comparison of the measured angular light scattering functions at 532 nm (data from first panel of Fig. 4) and theoretical phase functions for different habits calculated using the database of Yang et al. (2013) and assuming a size distribution as measured during the ACRIDICON-CHUVA campaign. All calculations…"*.

Figure 4 and Figure 5: are the measured 'angular light scattering functions' the same in both figures? If yes, please indicate. Also, roughened particles are used here for comparison because studies indicate that they perform well in many applications. How would the smooth particle model curve look like if they are overplotted in Figures 4 and 5?
The data in the first panel of Fig. 4 is the same as seen in Fig. 5. We indicated this in the Fig. 5 caption: *"A comparison of the measured angular light scattering functions at 532 nm (data from first panel of Fig. 4)…"*.

A comparison of phase functions of smooth and roughened particles from the used database are shown in Fig. 13 of Yang et al. (2013). This figure shows that smooth particles have distinct features in the phase function that are not well represented by our measurements. Overplotting the phase functions of smooth particles to Figs. 4 and 5 would make the figures too full and could

distract the reader from evaluating the difference between the measurements and the different severely roughened models (please see figure below). We think that the comparison between smooth and roughened particles is adequately discussed and shown in the cited literature so that it is not needed to show the smooth phase functions in Figs. 4 and 5. Further, the differences in the phase functions of smooth and roughened particles are described in the Introduction and in Sect. 4.

[Figure]

Line 8: ' the global mean change in the SWCRE is -1.12 W m-2', please indicate that more cooling is brought in using new g parameterization.
We have modified the first sentence of paragraph 4.2 to: "*The change in the global SWCRE after applying the new parameterization to all ice clouds (cirrus and mixed-phase) is shown in Fig. 8. The global mean change in the SWCRE is −1.12 W m$^{-2}$, but…*"

Line 9: ' the change in global SWCRE is small compared to ....', yes, it is right. However, based on Section 2.5, SWCRE is for ice clouds only. Is the change significant relative to your simulated SWCRE with new and old parameterizations? How about compare to SWCRE by ice clouds from [Gasparini and Lohmann, 2016] and [Hong et al., 2016] where show ice cloud radiative effect using ECHAM-HAM model and from observations?
The change in the SWCRE take into account all ice clouds (cirrus and mixed-phase). To make this more clear we modified the sentence on page 10 line 7 (old manuscript) the following: *"The change in the global SWCRE after applying the new parameterization to all ice clouds (cirrus and mixed-phase) is shown in…".*

We can directly compare our estimated change in the SWCRE to the one SWCRE calculated by Hong et al. (2016) but not to the one SWCRE calculated by Gasparine and Lohmann (2016) since this was calculated for cirrus clouds only. We have calculated the change in the SWCRE also only for cirrus clouds but this estimation was not shown in the manuscript. We added this information to the revised manuscript and modified the discussion the following: *"If the new parameterization is applied only for cirrus clouds, the mean change in the SWCRE is slightly lower, −1.00 W m$^{-2}$. Therefore, the change in the asymmetry factor mostly affects the cirrus SWCRE and, also, the largest effect is found in the tropical regions where also the cirrus occurrence is the highest (e.g. Sassen et al., 2008). Even though the change in the global SWCRE is small compared to the*

*global mean SWCRE of all clouds of about −50 W m$^{-2}$ (Boucher et al., 2013) or to the global mean SWCRE of ice clouds of about (−16.7± 1.7 W m$^{-2}$) (Hong et al., 2016) it is one fourth of the global mean cirrus SWCRE of −4 W m$^{-2}$ (Gasparini and Lohmann, 2016) and comparable to the total direct radiative effect of aerosols (−2.1± 0.7 W m$^{-2}$) (Lacagnina et al., 2017)".*

Line 11: ' the decrease in SWCRE...', please indicate cooling is enhanced.
We replaced the term *"the decrease in SWCRE"* with *"The enhanced SW cooling"*.

'cirrus CRE', please explain what ice clouds have been used for CRE studied? Thin cirrus only?
The term "cirrus CRE" here refers to the CRE by cold (<-40°C) ice clouds. Here, the term "cirrus CRE" is used to generally refer to all cirrus clouds.

**Authors' Response to Interactive Comment by Z Ulanowski**

The authors appreciate the Interactive Comments made by Z Ulanowski. Below we provide our answers to the raised issues. The Interactive Comments are in blue and authors' replies in black.

This extensive study investigates a very important area concerning the radiative impact of atmospheric ice. It could make an important contribution to this subject. However, several conclusions being made are too strong in my view and should be qualified. There is also one large flaw that should be addressed to increase the value of the study.
We thank Z Ulanowski for acknowledging the importance of this study and address his comments below.

4.2 p.10. My main point is a significant weakness of this study, the omission of long-wave (LW) effects of cirrus. To illustrate the importance of this shortcoming, the cirrus radiative effect difference found here is dominated by changes in the Tropical Warm Pool (TWP) and Maritime Continent. Yet in this region the net radiative influence of cirrus is determined largely by the longwave, with difference from even the zonal average of the order of many tens of W/mˆ2 (e.g. Xu and Guan, 2017; NOAA/ESRL), in contrast to the _peak_ SW value of about 8W/mˆ2 reported here. So potentially not just the magnitude but even the sign of the postulated effect could change. Hence the LW effect should be taken into account. The severely roughened hexagonal aggregate model that is adopted by the authors includes IR properties. Why were they not included to obtain the net radiative effect? Was the longwave parameterization done but the effects are not shown - why, it should be easy to do? Or was the parameterization not applied - which makes the model internally inconsistent? If this result is being kept "for later", I would strongly advise against it - salami-slicing climate science is a risky undertaking, e.g. the longwave cloud feedback is reported to be positive, mostly due to tropical cirrus (Zelinka and Hartmann, 2010), potentially negating the main conclusion from the work.
In this study we only discuss the effect of ice crystal complexity to the SWCRE and omitting the LW effect will not in any way change the conclusion of this work. There are two reasons why we do not discuss the LW effect. First, the focus of this study is the effect of ice crystal complexity on the ice cloud asymmetry factor. In the ECHAM-HAM model the ice particle asymmetry factors are only considered for calculation of the SW effect and are not included in the calculations of the LW effect. This is due to the fact that the LW effect is less sensitive to the ice crystal morphology than the SW effect. For example, Yi et al. (2013) showed that changing the ice crystals from smooth to complex will not significantly affect the LWCRE.

Secondly, our optical measurements are in the SW region and, therefore, we can only make conclusion of the SW asymmetry factors. We agree that optical measurements in the LW region would be of interest to validate LW parameterizations in the future. Furthermore, we think that the term "salami-slicing" is more than misplaced with regard to this work, which - in our opinion - represents one of the most comprehensive studies on ice crystal complexity and its influence on the cloud radiative forcing. The experimental data used here are from dedicated cloud chamber simulation runs as well as from the field, gathered in a dozen of aircraft projects around the globe. Further, the data are used to construct a new, more realistic parameterization of the asymmetry factor to be used in climate models - a scientific span that is not common in the field.

This brings me to a related point: the authors make strong statements about the radiative impact, with the largest impact being demonstrated in the TWP/MC region. Yet no in situ data from this region is provided, and very little data from the tropics altogether. What there is, refers to Amazonia, where modelling indicates very weak impact.
The data presented in this study covers all the geographical areas where the KIT SID-3 and the PHIPS instruments have been flown and a large amount of the campaigns where PN measurements were available. We agree that the TWP/MC region (where these instruments have not yet flown) is highly important for the cirrus cloud radiative impact. We hope that in future more field campaigns will be focused on this area, where this study demonstrates the largest SW impact.

Some smaller points follow.

Introduction p.2 and section 2.1 p.3. I find it surprising that the authors do not properly acknowledge that SID3, the core instrument in this work, and long-term assistance with the hardware, software and data analysis techniques were provided to KIT by the team at University of Hertfordshire.

The SID-3 instrument was developed by the University of Hertfordshire and a version of the instrument was purchased by KIT in 2008. It is true that many collaborative efforts between KIT and Hertfordshire has taken place to improve the hardware, software and data analysis methods to the current status and we believe that these collaborative efforts are correctly documented in the corresponding literature. The instrument itself is cited through the original Hertfordshire publication of Kaye et al. (2008). The University of Hertfordshire was involved in the first field deployment of the instrument in the MACPEX campaign, which is acknowledged by co-authorship in the Järvinen et al. (2016) and Schmitt et al. (2016b) publications. The SID-3 scattering pattern analysis methods for atmospheric ice particles were also developed in close collaboration between KIT an University of Hertfordshire by conducting at least five joint AIDA cloud simulation campaigns. This effort is acknowledged in the original work describing the use of the complexity parameter, $k_e$, as a complexity measure (Schnaiter et al., 2016), where the University of Hertfordshire (Z Ulanowski) is listed as co-author.

2.1 p.3. Likewise, the method for determining ice crystal roughness using pattern texture analysis (including GLCM) was developed by the Hertfordshire group (Ulanowski et al., 2010, 2014). This should be acknowledged too.

Analysing scattering patterns to retrieve information on surface roughness has been previously used in industrial applications for surface quality control (e.g. Lu et al., 2006) but it is true that the Hertfordshire group was the first to use this technique for ice crystal surface roughness. Therefore, we have added the citation to Ulanowski et al., 2010, 2014 to the following sentence: "*The crystal complexity is quantified from the 2-D scattering patterns using a grey-level co-occurrence matrix (GLCM) method (Lu et al., 2006). This method was developed for industrial quality control of surface treatment processes but was later adapted for analysis of complexity features of three-dimensional ice particles (Ulanowski et al., 2010, 2014; Schnaiter et al., 2016).*"

3.2 p.7. "enhanced submicron scale complexity of homogeneously formed ice crystals [...] and can be explained by an increased stacking disorder of homogeneously nucleated ice crystals" Firstly, it would be difficult to associate in situ measurements with the homogeneous mode of nucleation in such categorical fashion. The second part of this statement is extremely simplistic too, no proof of a general connection of complexity with stacking disorder exists yet, even in the lab let alone the atmosphere. While stacking-disordered ice has been produced in the supercooled water freezing experiments of Malkin et al. (2012), heterogeneous ice nucleation is equally important and there can be other reasons why roughness arises (Chou et al., 2018).

We refer to the in situ measurements that were presented in Ulanowski et al. (2014). The authors argued that in situ observations in a mid-latitudes cirrus showed differences in the ice crystal complexity based on the airmass origin: "*polluted airflow showed significantly lower roughness for all measures apart from kurtosis. We speculate that this was due to higher concentration of inhomogeneous ice nuclei (IN) in the last case*". Of course it is difficult to investigate the origin of ice crystal complexity based on in situ measurements, especially if the ice particle history is unknown. Therefore, such laboratory studies will be valuable to interpreted in situ field results.

For the second point, we agree that our knowledge of formation of surface roughness in a single crystal is still highly unknown. Therefore, we modified the sentence as: "*can be partly explained by…*".

3.2. p7. While cyclic growth has been shown to contribute to increased ice roughness (Chou et al., 2018) the SEM experiments that are cited (Magee et al., 2014) are thought to have limited relevance to ice behaviour at tropospheric conditions, as growth in the near-vacuum of a SEM takes place under kinetically-limited, not diffusion-limited conditions typical of the troposphere (Kiselev, 2014; Chou et al., 2018).

We agree that discussing the results of SEM experiments in atmospheric context is challenging due to the near-vacuum pressure conditions experienced by the ice crystals. Therefore, it is important to have proof such results in atmospheric conditions as shown in Chou et al. (2018). We have added this reference to the sentence together with the Magee et al. (2014) reference.

[revised manuscript text omitted]

15 ~~ice particle would produce speckles in a similar way as would an ice particle with air inclusions. Therefore, we refrain from using the more established term surface roughness and instead use the term submicron scale complexity that includes not only surface roughness but also all the other possible causes for the speckles in the 2-D scattering patterns, such as hollowness or air inclusions. In previous studies, such complexity has been referred as the small-scale complexity (Schnaiter et al., 2016; Baran et al., 2017). Here, we suggest to replace the term small with submicron since it is known that the scale of this complexity is in the submicron~~
20

[revised manuscript text omitted]
 less than that of pristine ice crystals (Yang et al., 2013). To estimate the asymmetry factor from our measurements, the measured light scattering functions need to be extrapolated. Our approach was to find a best fit to the angular light scattering measurements using a theoretical phase function whose asymmetry factor is knownsize distribution for ice particlesdiameterFig. 5 showsand the theoretical angular scattering functions calculated based on the scattering data base of Yang et al. (2013) and assuming severely roughened particles~~for 532 and $804\,nm$ and the theoretical phase functions for nine different habits.

Based on the comparison, the severely roughened column aggregate model was found to best represent the measurement at both wavelengths. At $532\,nm$ the theoretical calculations agree with the measurements over the whole measurement range, whereas at $804\,nm$ the model predicts slightly higher intensity in the sideward angles between $57°$ and $126°$ but is within the measured interquartile range (Fig. 6). The calculated root mean square errors (RMSE) between the severely roughened column aggregate model and the mean of the measurements are the lowest (0.0017 and 0.0014 for 532  and 804 nm, respectively) compared to the other models (RMSEs between 0.0022 and 0.0111 for 532 nm, and 0.0037 and 0.0208 for 804 nm). At the angles around exact-backscattering the severely roughened column aggregate model predicts a relatively flat

behaviour. However, recent modelling studies have indicated that the scattering intensities around exact backscattering angles should be enhanced due to coherent scattering (e.g. Zhou, 2018). Although this effect can be important for lidar applications, it does not significantly affect the redistribution of the energy in the scattering process and, thus, the magnitude of the asymmetry factor. Furthermore, comparisons of satellite retrievals of cloud polarization properties with light scattering simulations have

5    shown that optical particle models using severely roughened crystals yield the best agreement (Baum et al., 2011; Yang et al., 2013; Tang et al., 2017) and the current MODIS retrievals are based on the same optical particle model of severely roughened hexagonal aggregates that is used here (Platnick et al., 2017). At 804 nm the model predicted slightly higher intensity in the sideward angles between 57° and 126° but was within the measured interquartile range (Fig. 3). Using

**4    Estimating the effect of the observed mesoscopic scale complexity to SWCRE**

10    An important consequence of severely roughened and complex ice crystals is that the cloud asymmetry factor in the solar spectral range is lowered compared to pristine ice crystals (e.g. Macke et al., 1996; Yang and Liou, 1998; Liou et al., 2000; Baum et al., 20 For example, the severely roughened hexagonal aggregate model has relatively low asymmetry factors of 0.750 and 0.754 at 532 nm and 804 nm, respectively, were retrieved.

**5    Estimating the effect of the observed submicron scale complexity to SWCRE**

15    for 532 nm and 804 nm, respectively. To understand the relevance of our observations for climate projections, the effect of the observed decrease in the cloud asymmetry parameter on the SWCRE was estimated . The measured angular light scattering function was used to derive a new parameterization of by newly parameterizing the SW asymmetry factors and this paramererization was tested using the optical model with the best fit to our measurements in the ECHAM-HAM global climate model. The current optical parameterization in the ECHAM-HAM model is calculated based on spherical particles using

20    Mie-theory with the exception that the asymmetry factors are scaled down to be more representative for aspherical ice particles. The steps to retrieve this parametrization are sdiscussed the new parametrization are discussed in Sect. 4.1. The sensitivity of a global climate model to the ice particle surface roughness has already been tested in the study of Yi et al. (2013), where the difference in the SWCRE was calculated for assuming first completely smooth and later severely roughened ice particles. Here, we compare the existing standard parameterization of SW asymmetry factors to our new parametrerization and, in this

25    way, estimate the uncertainty in the SWCRE due to the failure to adequately consider ice crystal submicron scale complexity.

**4.1    Derivation of the new parameterization of the short-wave asymmetry factor for the ECHAM-HAM model and comparison with the standard parameterization.**

Parameterizations of the ice crystal shortwave radiative properties are frequently based on possible impact of of the observed ice crystal habits and size distributions (e.g. Wyser and Yang, 1998; Yang et al., 2000; McFarquhar et al., 2002; Um and McFarquhar, 2007) w

30    the assumption that a link exists between the ice crystal microphysical and bulk scattering properties. McFarquhar et al. (2002) showed

that different parameterizations using different habits or habit mixtures can cause a significant variance in the asymmetry factor by 0.07 in the wavelength band of 0.25 to 0.69 μm. This variance becomes especially significant for small ice particles, with effective radius below 20 μm, where also the largest uncertainty in the exact particle form exists. Most small ice particles are classified as quasi-spherical particles based on in-situ cloud particle imaging measurements (McFarquhar and Heymsfield, 1996; Stith et al.

5  however angular light scattering measurements have shown that the small quasi-spherical particles can have very different asymmetry factors than that of a spherical particle depending on the degree of submicron complexity (Järvinen et al., 2016). mesoscopic scale complexity to the SWCRE.

To overcome the problem of relating the ice crystal bulk scattering properties to measured microphysical properties, in this study a new parameterization of the SW asymmetry factors is developed using direct measurements of the bulk optical

10  properties.

[revised manuscript text omitted]

Voigt, C., Schumann, U., Minikin, A., Abdelmonem, A., Afchine, A., Borrmann, S., Boettcher, M., Buchholz, B., Bugliaro, L., Costa, A., Curtius, J., Dollner, M., Dörnbrack, A., Dreiling, V., Ebert, V., Ehrlich, A., Fix, A., Forster, L., Frank, F., Fütterer, D., Giez, A., Graf, K., Grooß, J.-U., Groß, S., Heinold, B., Hüneke, T., Järvinen, E., Jurkat, T., Kaufmann, S., Kenntner, M., Klingebiel, M., Klimach, T., Kohl, R., Krämer, M., Krisna, T. C., Luebke, A., Mayer, B., Mertes, S., Molleker, S., Petzold, A., Pfeilsticker, K., Port, M., Rapp, M., Reutter, P., Rolf, C., Rose, D., Sauer, D., Schäfler, A., Schlage, R., Schnaiter, M., Schneider, J., Spelten, N., Spichtinger, P., Stock, P., Weigel, R., Weinzierl, B., Wendisch, M., Werner, F., Wernli, H., Wirth, M., Zahn, A., Ziereis, H., and Zöger, M.: ML-CIRRUS - The airborne experiment on natural cirrus and contrail cirrus with the high-altitude long-range research aircraft HALO, Bull. Amer. Meteor. Soc., 98, 271–288, 2017.

Wagner, R., Möhler, O., Saathoff, H., Schnaiter, M., and Leisner, T.: New cloud chamber experiments on the heterogeneous ice nucleation ability of oxalic acid in the immersion mode, Atmos. Chem. Phys., 11, 2083–2110, 2011.

Wendisch, M. and et al.: The Arctic Cloud Puzzle: Using ACLOUD/PASCAL Multi-Platform Observations to Unravel the Role of Clouds and Aerosol Particles in Arctic Amplification, Submitted to Bull. Am. Meteorol. Soc., 2018.

Wendisch, M., Pöschl, U., Andreae, M. O., Machado, L. A., Albrecht, R., Schlager, H., Rosenfeld, D., Martin, S. T., Abdelmonem, A., Afchine, A., et al.: The ACRIDICON-CHUVA campaign: Studying tropical deep convective clouds and precipitation over Amazonia using the new German research aircraft HALO, Bull. Amer. Meteor. Soc., 97, 1885–1908, 2016.

Wyser, K. and Yang, P.: Average ice crystal size and bulk short-wave single-scattering properties of cirrus clouds, Atmospheric research, 49, 315–335, 1998.

Yang, P. and Liou, K.: Single-scattering properties of complex ice crystals in terrestrial atmosphere, Beitrage zur Physik der Atmosphare-Contributions to Atmospheric Physics, 71, 223–248, 1998.

Yang, P., Liou, K., Wyser, K., and Mitchell, D.: Parameterization of the scattering and absorption properties of individual ice crystals, Journal of Geophysical Research: Atmospheres, 105, 4699–4718, 2000.

[revised manuscript text omitted]